# DeepAveragers: Offline Reinforcement Learning by Solving Derived Non-Parametric MDPs

**Aayam Shrestha, Stefan Lee, Prasad Tadepalli, Alan Fern**
Oregon State University
Corvallis, OR 97330, USA
`{shrestaa, leestef, tadepall, alan.fern}@oregonstate.edu`

## Abstract

We study an approach to offline reinforcement learning (RL) based on optimally solving finitely-represented MDPs derived from a static dataset of experience. This approach can be applied on top of any learned representation and has the potential to easily support multiple solution objectives as well as zero-shot adjustment to changing environments and goals. Our main contribution is to introduce the Deep Averagers with Costs MDP (DAC-MDP) and to investigate its solutions for offline RL. DAC-MDPs are a non-parametric model that can leverage deep representations and account for limited data by introducing costs for exploiting under-represented parts of the model. In theory, we show conditions that allow for lower-bounding the performance of DAC-MDP solutions. We also investigate the empirical behavior in a number of environments, including those with image-based observations. Overall, the experiments demonstrate that the framework can work in practice and scale to large complex offline RL problems.

## 1 Introduction

Research in automated planning and control has produced powerful algorithms to solve for optimal, or near-optimal, decisions given accurate environment models. Examples include the classic value- and policy-iteration algorithms for tabular representations or more sophisticated symbolic variants for graphical model representations (e.g. Boutilier et al. (2000); Raghavan et al. (2012)). In concept, these planners address many of the traditional challenges in reinforcement learning (RL). They can perform "zero-shot transfer" to new goals and changes to the environment model, accurately account for sparse reward or low-probability events, and solve for different optimization objectives (e.g. robustness). Effectively leveraging these planners, however, requires an accurate model grounded in observations and expressed in the planner's representation. On the other hand, model-based reinforcement learning (MBRL) aims to learn grounded models to improve RL's data efficiency. Despite developing grounded environment models, the vast majority of current MBRL approaches do not leverage near-optimal planners to help address the above challenges. Rather, the models are used as black-box simulators for experience augmentation and/or Monte-Carlo search. Alternatively, model learning is sometimes treated as purely an auxiliary task to support representation learning.

The high-level goal of this paper is to move toward MBRL approaches that can effectively leverage near-optimal planners for improved data efficiency and flexibility in complex environments. However, there are at least two significant challenges. First, there is a mismatch between the deep model representations typically learned in MBRL (e.g. continuous state mappings) and the representations assumed by many planners (e.g. discrete tables or graphical models). Second, near-optimal planners are well-known for exploiting model inaccuracies in ways that hurt performance in the real environment, e.g. (Atkeson, 1998). This second challenge is particularly significant for offline RL, where the training experience for model learning is fixed and limited.

We address the first challenge above by focusing on tabular representations, which are perhaps the simplest, but most universal representation for optimal planning. Our main contribution is an offline MBRL approach based on optimally solving a new model called the Deep Averagers with Costs MDP (DAC-MDP). A DAC-MDP is a non-parametric model derived from an experience dataset and a corresponding (possibly learned) latent state representation. While the DAC-MDP is defined over the entire continuous latent state space, its full optimal policy can be computed by solving a standard (finite) tabular MDP derived from the dataset. This supports optimal planning via any

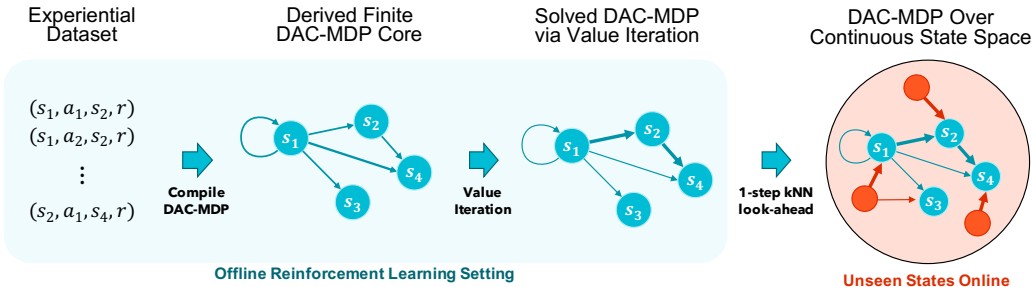

Figure 1: Overview of Offline RL via DAC-MDPs. Given a static experience dataset, we first compile it into a finite tabular MDP which is at most the size of the dataset. This MDP contains the "core" states of the full continuous DAC-MDP. The finite core-state MDP is then solved via value iteration, resulting in a policy and Q-value function for the core states. This finite Q-function is used to define a non-parametric Q-function for the continuous DAC-MDP, which allows for Q-values and hence a policy to be computed for previously unseen states.

tabular MDP solver, e.g. value iteration. To scale this approach to typical offline RL problems, we develop a simple GPU implementation of value iteration that scales to millions of states. As an additional engineering contribution, this implementation will be made public.

To address the second challenge of model inaccuracy due to limited data, DAC-MDPs follow the *pessimism in the face of uncertainty* principle, which has been shown effective in a number of prior contexts (e.g. (Fonteneau et al., 2013)). In particular, DAC-MDPs extend Gordon's Averagers framework (Gordon, 1995) with additional costs for exploiting transitions that are under-represented in the data. Our second contribution is to give a theoretical analysis of this model, which provides conditions under which a DAC-MDP solution will perform near optimally in the real environment.

Our final contribution is to empirically investigate the DAC-MDP approach using simple latent representations derived from random projections and those learned by Q-iteration algorithms. Among other results, we demonstrate the ability to scale to Atari-scale problems, which is the first demonstration of optimal planning being effectively applied across multiple Atari games. In addition, we provide case studies in 3D first-person navigation that demonstrate the flexibility and adaptability afforded by integrating optimal planning into offline MBRL. These results show the promise of our approach for marrying advances in representation learning with optimal planning.

## 2  FORMAL PRELIMINARIES

A Markov Decision Process (MDP) is a tuple $\langle \mathcal{S}, \mathcal{A}, T, R \rangle$ (Puterman, 1994), with state set $\mathcal{S}$, action set $\mathcal{A}$, transition function $T(s, a, s')$, and reward function $R(s, a)$. A policy $\pi$ maps states to actions and has Q-function $Q^\pi(s, a)$ giving the expected infinite-horizon $\beta$-discounted reward of following $\pi$ after taking action $a$ in $s$. The optimal policy $\pi^*$ maximizes the Q-function over all policies and state-action pairs. $Q^*$ corresponds to the optimal Q-function that satisfies $\pi^*(s) = \arg\max_a Q^*(s, a)$. $Q^*$ can be computed given the MDP by repeated application of the *Bellman Backup Operator* $B$, which for any Q-function $Q$, returns a new Q-function given by,

$$B[Q](s, a) = R(s, a) + \gamma \mathbb{E}_{s' \sim T(s, a, \cdot)} \left[ \max_a Q(s', a) \right]. \tag{1}$$

The objective of RL is to find a near-optimal policy without prior knowledge of the MDP. In the online RL setting, this is done by actively exploring actions in the environment. Rather, in the offline RL (Levine et al., 2020), which is the focus of this paper, learning is based on a static dataset $\mathcal{D} = \{(s_i, a_i, r_i, s'_i)\}$, where each tuple gives the reward $r_i$ and next state $s'_i$ observed after taking action $a_i$ in state $s_i$.

In strict offline RL setting, the final policy selection must be done using only the dataset, without direct access to the environment. This includes all hyperparameter tuning and the choice of when to stop learning. Evaluations of offline RL, however, often blur this distinction, for example, reporting performance of the best policy obtained across various hyperparameter settings as evaluated via new online experiences (Gulcehre et al., 2020). Here we consider an evaluation protocol that makes the amount of online access to the environment explicit. In particular, the offline RL algorithm is

allowed to use the environment to evaluate $N_e$ policies (e.g. an average over repeated trials for each policy), which, for example, may derive from different hyperparameter choices. The best of the evaluated policies can then be selected. Note that $N_e = 1$ corresponds to pure offline RL.

## 3 DEEP AVERAGERS WITH COSTS MDPS (DAC-MDPS)

From a practical perspective our approach carries out the following steps as illustrated in Figure 1. (1) We start with a static experience dataset, where the states are assumed to come from a continuous latent state space. For example, states encoded via random or learned deep representations. (2) Next we compile the dataset into a tabular MDP over the "core states" of the DAC-MDP (those in the dataset). This compilation uses $k$-nearest neighbor (kNN) queries to define the reward and transition functions (Equation 2) functions of the core states. (3) Next we use a GPU implementation of value iteration to solve for the tabular MDP's optimal Q-function. (4) Finally, this tabular Q-function is used to define the Q-function over the entire DAC-MDP (Equation 3). Previously unseen states at test time are assigned Q-values and in turn a policy action via kNN queries over the core states.

Conceptually, our DAC-MDP model is inspired by Gordon's (1995) early work that showed the convergence of (offline) approximate value iteration for a class of function approximators; *averagers*, which includes methods such as $k$ nearest neighbor (kNN) regression, among others. It was also observed that approximate value iteration using an averager was equivalent to solving an MDP derived from the offline data. That observation, however, was not investigated experimentally and has yet to be integrated with deep representation learning. Here we develop and evaluate such an integration.

The quality of an averagers MDP, and model-learning in general, depends on the size and distribution of the dataset. In particular, an optimal planner can exploit inaccuracies in the underrepresented parts of the state-action space, which can lead to poor performance in the real environment. The DAC-MDPs aim to avoid this by augmenting the derived MDPs with costs/penalties on under-represented transitions. This turns out to be essential to achieving good performance on challenging benchmarks.

### 3.1 DAC-MDP DEFINITION

A DAC-MDP is defined in terms of an experience dataset $\mathcal{D} = \{(s_i, a_i, r_i, s_i')\}$ from the true MDP $\mathcal{M}$ with continuous latent state space $\mathcal{S}$ and finite action space $\mathcal{A}$. The DAC-MDP $\tilde{M} = (\mathcal{S}, \mathcal{A}, \tilde{R}, \tilde{T})$ shares the same state and action spaces as $\mathcal{M}$, but defines the reward and transition functions in terms of empirical averages over the $k$ nearest neighbors of $(s, a)$ in $\mathcal{D}$.

The distance metric $d(s, a, s', a')$ gives the distance between pairs $(s, a)$ and $(s', a')$. This metric considers (s,a) pairs with different actions to be *infinitely distant*. Otherwise, the distance between pairs involving the same action is the *euclidean distance* between their states. In particular, the distance between $(s, a)$ and a data tuple $(s_i, a_i, r_i, s_i')$ is given by $d(s, a, s_i, a_i)$. Also, we let $kNN(s, a)$ denote the set of indices of the $k$ nearest neighbors to $(s, a)$ in $\mathcal{D}$, noting that the dependence on $\mathcal{D}$ and $d$ is left implicit. Given hyperparameters $k$ (smoothing factor) and $C$ (cost factor) we can now specify the DAC-MDP reward and transition function.

$$\tilde{R}(s, a) = \frac{1}{k} \sum_{i \in kNN(s,a)} r_i - C \cdot d(s, a, s_i, a_i), \quad \tilde{T}(s, a, s') = \frac{1}{k} \sum_{i \in kNN(s,a)} I[s' = s_i'] \quad (2)$$

The reward for $(s, a)$ is simply the average reward of the nearest neighbors with a penalty for each neighbor that grows linearly with the distance to a neighbor. Thus, the farther $(s, a)$ is to its nearest neighbor set, the less desirable its immediate reward will be. The transition function is simply the empirical distribution over destination states of the nearest neighbor set.

Importantly, even though a DAC-MDP has an infinite continuous state space, it has a special finite structure. Since the transition function $\tilde{T}$ only allows transitions to states appearing as destination states in $\mathcal{D}$. We can view $\tilde{M}$ as having a finite core set of states $\mathcal{S}_D = \{s_i' \mid (s_i, a_i, r_i, s_i') \in \mathcal{D}\}$. States in this core do not transition to non-core states and each non-core state immediately transitions to the core for any action. Hence, the value of core states is not influenced by non-core states. Further, once the core values are known, we can compute the values of any non-core state via one-step look ahead using $\tilde{T}$. *Thus, we can optimally solve a DAC-MDP by solving just its finite core.*

Specifically, let $\tilde{Q}$ be the optimal Q-function of $\tilde{M}$. We can compute $\tilde{Q}$ for the core states by solving the finite MDP $\tilde{M}_D = (\mathcal{S}_D, \mathcal{A}, \tilde{R}, \tilde{T})$. We can then compute $\tilde{Q}$ for any non-core state on demand

via the following one-step look-ahead expression.[1] This allows us to compute the optimal policy of $\tilde{M}$, denoted $\tilde{\pi}$, using any solver of finite MDPs.

$$\tilde{Q}(s,a) = \frac{1}{k} \sum_{i \in kNN(s,a)} r_i + \gamma \max_a \tilde{Q}(s_i',a) - C \cdot d(s,a,s_i,a_i) \tag{3}$$

## 3.2 DAC-MDP Performance Lower Bound

We are ultimately interested in how well the optimal DAC-MDP $\tilde{M}$ policy $\tilde{\pi}$ performs in the true MDP $\mathcal{M}$. Without further assumptions, $\tilde{\pi}$ can be arbitrarily sub-optimal in $\mathcal{M}$, due to potential "non-smoothness" of values, limited data, and limited neighborhood sizes. We now provide a lower-bound on the performance of $\tilde{\pi}$ in $\mathcal{M}$ that quantifies the dependence on these quantities. Smoothness is characterized via a Lipschitz smoothness assumptions on $B[\tilde{Q}]$, where $B$ is the Bellman operator for the true MDP and $\tilde{Q}$ is the optimal Q-function for $\tilde{M}$ using hyperparameters $k$ and $C$. In particular, we assume that there is a constant $L(k,C)$, such that for any state-action pairs $(s,a)$ and $(s',a')$

$$\left| B[\tilde{Q}](s,a) - B[\tilde{Q}](s',a') \right| \leq L(k,C) \cdot d(s,a,s',a').$$

This quantifies the smoothness of the Q-function obtained via one-step look-ahead using the true dynamics The coverage of the dataset is quantified in terms of the worst case average distance to a kNN set defined by $\bar{d}_{max} = \max_{(s,a)} \frac{1}{k} \sum_{i \in kNN(s,a)} d(s,a,s_i,a_i)$. The bound also depends on $Q_{max} = \max_{(s,a)} \tilde{Q}(s,a)$ and $M_{N,k}$, which is the maximum number of distinct kNN sets over a dataset of size $N$. For $L2$ distance over a $d$-dimensional space, $M_{N,k}$ is bounded by $O\left( (kN)^{\frac{d}{2}+1} \right)$.

**Theorem 3.1.** *For any data set $\mathcal{D}$ of size $N$, let $\tilde{Q}$ and $\tilde{\pi}$ be the optimal Q-function and policy for the corresponding DAC-MDP with parameters $k$ and $C$. If $B[\tilde{Q}]$ is Lipshitz continuous with constant $L(k,C)$, then with probability at least $1 - \delta$,*

$$V^{\tilde{\pi}} \geq V^* - \frac{2\left( L(k,C) \cdot \bar{d}_{max} + Q_{max}\epsilon(k,N,\delta) \right)}{1-\gamma}, \quad \epsilon(k,N,\delta) = \sqrt{\frac{1}{2k} \ln \frac{2M_{N,k}}{\delta}},$$

*which for $L2$ distance over a $d$-dimensional space yields $\epsilon(k,N,\delta) = O\left( \sqrt{\frac{1}{k}\left( d\ln kN + \ln \frac{1}{\delta} \right)} \right)$.*

*The full proof is in the Appendix A.2.* The first term in the bound characterizes how well the dataset represents the MDP from a nearest neighbor perspective. In general, $\bar{d}_{max}$ decreases as the dataset becomes larger. The second term characterizes the variance of stochastic state transitions in the dataset and decreases with the smoothness parameter $k$. Counter-intuitively we see that for Euclidean distance, the second term can grow logarithmically in the dataset size. This worst-case behavior is due to not making assumptions about the distribution of source states-actions in the dataset. Thus, we can't rule out adversarial choices that negatively influence the kNN sets of critical states.

**Practical Issues**: There are some key practical issues of the framework regarding the scalability of VI and selection of hyperparameters. We exploit the fixed sparse structure of DAC-MDPs along with parallel compute afforded by GPUs to gracefully scale our VI to millions of states. This can provide anywhere between 20-1000x wall clock speedup over its serial counterpart. To reduce the sensitivity to the smoothness parameter k, we use weighted averaging based on distances to nearest neighbors instead of uniform averaging in Equation 2. Furthermore, we use different smoothness parameter k and $k_\pi$ to build the DAC-MDP and compute the policy respectively. We find that using larger values of $k_\pi$ is generally beneficial to reduce the variance. Finally we use a simple rule-of-thumb of setting cost factor $C$ to be in the order of magnitude of observed rewards. These choices are further elaborated in detail in appendix A.1

## 3.3 Representation Learning

While DAC-MDPs can be defined over raw observations, for complicated observation spaces, such as images, performance will likely be poor in practice for simple distance functions. Thus, combining the DAC-MDP framework with deep representation learning is critical for observation-rich

---

[1]Note that we can get $\tilde{V}$ using value iteration on $\tilde{M}_D$. $\tilde{Q}$ can then be computed via 1-step lookup as $\tilde{Q}(s,a) = \tilde{R}(s,a) + \gamma \sum_{s' \in \tilde{T}(s,a)} \tilde{T}(s,a,s') \cdot \tilde{V}(s)$

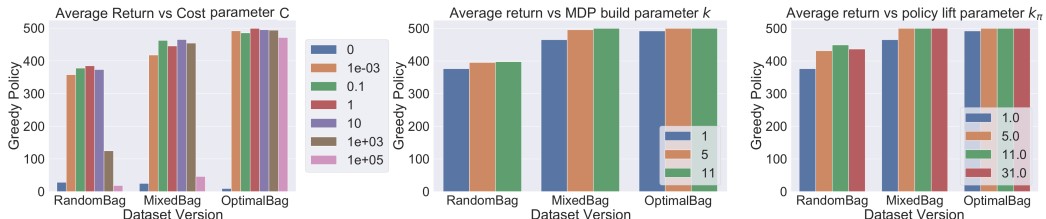

Figure 2: (a) Greedy Policy performance for CartPole with varying (a) cost paramter $C$. (b) smoothness parameter $k$. (c) policy smoothing parameter $k_\pi$

environments. Since the primary focus of this paper is on the introduction of DAC-MDPs, rather than representation learning, below we describe three basic representation-learning approaches used in our experiments. An important direction for future work is to investigate the effectiveness of other alternatives and to develop representation-learning specifically for the DAC-MDP framework.

**Random Projection** ($RND$): This baseline simply produces a representation vector as the output of a chosen network architecture with randomly initialized weights (e.g. a CNN with random weights for images). **Offline DQN** ($DQN$): From the first term of Theorem 3.1 we see that decreasing the Lipschitz constant of $B[\tilde{Q}]$ can improve the performance bound. This suggests that a representation that is smooth with respect to the Q-function can be beneficial. For this purpose, our second representation directly uses the penultimate layer of DQN (Mnih et al., 2013) after it has been trained on our offline dataset $\mathcal{D}$. Since the penultimate layer is just a linear mapping away from the DQN Q-values, it is reasonable to believe that this representation will be smooth with respect to meaningful Q-functions for the environment. **BCQ - Imitation** ($BCQ$): The offline RL algorithm BCQ (Fujimoto et al., 2019) trains an imitation network to emulate the action distribution of the behavioral policy. This policy is used to help avoid backing up action values for under-represented actions. We use the penultimate layer of the imitation network trained on our dataset as the third type of representation. This representation is not as well-motivated by our performance bounds, but is intuitively appealing and provides a policy-centric contrast to Q-value modeling.

## 4 EXPERIMENTS

The experiments are divided into three sections. First, we present exploratory experiments in Cartpole to illustrate the impact of DAC-MDP parameters using an idealized representation. Second, we demonstrate the scalability of the approach on Atari games with image-based observations. Finally, we demonstrate use-cases enabled by our planning-centric approach in a 3D first-person domain.

**Exploring DAC-MDP Parameters**: We explore the impact of the DAC-MDP parameters in Cartpole. To separate the impact of state-representation and DAC-MDP parameters, we use the ideal state representation consisting of the true 4-dimensional system state. We generate three datasets of size 100k each: (1) $RandomBag$ ($V^{\pi_\beta} = 20$).[2]: Trajectories collected by random policies. (2) $OptimalBag$ ($V^{\pi_\beta} = 140$): Trajectories collected by a well-trained DQN policy. (3) $MixedBag$ ($V^{\pi_\beta} = 500$): Trajectories collected by a mixed bag of $\epsilon$-greedy DQN policies using $\epsilon \in [0, 0.1, 0.2, 0.4, 0.6, 1]$.

We first illustrate the impact of the cost parameter $C$ by fixing $k = k_\pi = 1$ and varying $C$ from 0 to 1e6. Figure 2a shows that for all datasets when there is no penalty for under-represented transitions, i.e. $C = 0$ , the policy is extremely poor. This shows that even for this relatively simple problem with a relatively large dataset, the MDPs from the original cost-free averagers framework, can yield low-quality policies. At the other extreme, when $C$ is very large, the optimal policy tries to stay as close as possible to transitions in the dataset. This results in good performance for the *OptimalBag* dataset, since all actions are near optimal. However, the policies resulting from the *MixedBag* and *RandomBag* datasets fail. This is due to those datasets containing a large number of sub-optimal actions, which the policy should actually avoid for purposes of maximizing reward. Between these extremes the performance is relatively insensitive to the choice of $C$.

Figure 2b explores the impact of varying $k$ using $k_\pi = 1$ and $C = 1$. The main observation is that there is a slight disadvantage to using $k = 1$, which defines a deterministic finite MDP, compared to $k > 1$, especially for the non-optimal datasets. This indicates that the optimal planner is able

---

[2]$V^{\pi_\beta}$ is the expected performance of the behavioral policy for the dataset.

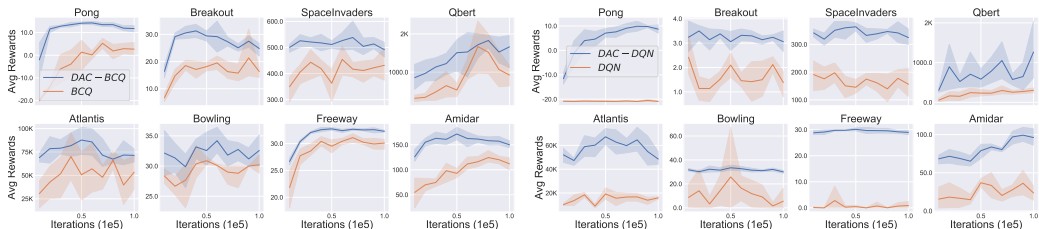

Figure 3: Results on Atari 100K (left)BCQ (right) DQN. Each agent is trained for 100K iterations(training steps), and evaluated on 10 episodes every 10K steps. At each of these evaluation checkpoints, we use the internal representation to compile DAC-MDPs. We then evaluate the DAC-MDPs for $N_e = 6$. Runs averaged over 5 seeds and error bars plot the 95% confidence interval.

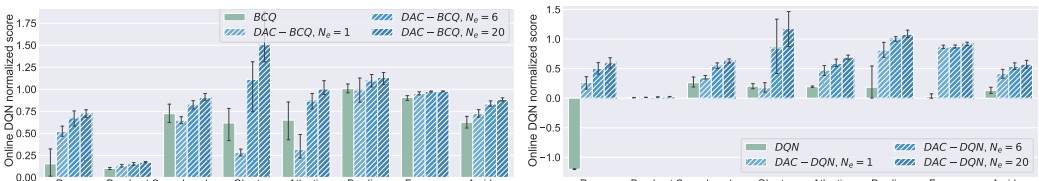

Figure 4: Results for different sets of candidate policies $N_e$ on 100K dataset. Here we plot the final performance of BCQ representation (left) and DQN representation (right) along with the DAC-MDP performances for different values of $N_e$. Runs averaged over 5 seeds. Error bars plot the 95% confidence interval.

to benefit by reasoning about the stochastic transitions of the DAC-MDP for $k > 1$. Otherwise the results are relatively insensitive to $k$. We set $k = 5$ for remaining experiments unless otherwise stated. Finally, Figure 2c varies the policy smoothing parameter $k_\pi$ from 1 to 31. Similar to the results for varying $k$, there is a benefit to using a value of $k_\pi > 1$, but otherwise the sensitivity is low. Thus, even for this relatively benign problem setting, some amount of averaging at test time appears beneficial. We find that DAC-MDPs are slightly more sensitive to these parameters in low data regime. Similar experiments for low data regime are presented in Appendix A.4.

**Experiments on Atari**: We choose 8 stochastic Atari games ranging from small to medium action spaces: Pong, Breakout, Space Invaders, Q Bert, Atlantis, Bowling, Freeway, and Amidar. These are stochastic in nature as sticky actions have been enabled. Following recent work (Fujimoto et al., 2019), we generate datasets by first training a DQN agent for each game. Next, to illustrate performance on small datasets and scalability to larger datasets, we generated two datasets of sizes 100K and 2.5 million for each game using an $\epsilon$-greedy version of the DQN policy. In particular, each episode had a 0.8 probability of using $\epsilon = 0.2$ and $\epsilon = 0.001$ otherwise. This ensures a mixture of both near optimal trajectories and more explorative trajectories.

We first trained offline DQN and BCQ agents on the 100K dataset for 100K iterations and evaluated on 10 test episodes every 10K iterations. At each evaluation iteration, we construct DAC-MDPs using the latent state representation at that point. We consider an offline RL setting where $N_e$=6, i.e. we evaluate 6 policies (on 10 episodes each) corresponding to 6 different DAC-MDPs, set to a parameter combination of $k$=5, $C \in \{1, 100, 1M\}$, and $k_\pi \in \{11, 51\}$. For each representation and evalutaion point, the best performing DAC-MDP is recorded. The entire 100K iteration protocol was repeated 3 times. Figure 3 shows the averaged curves with 90% confidence intervals.

The main observation is that for both BCQ and DQN representations the corresponding DAC-MDP performance (labeled DAC-BCQ and DAC-DQN) is usually better than BCQ or DQN policy at each point. Further, the DAC-MDP performance is usually better than the maximum performance of pure BCQ/DQN across the iterations. These results point to an interesting possibility of using DAC-MDPs to select the policy at the end of training at any fixed number of training iterations, rather than using online evaluations along the curve. Interestingly, the first point on each curve corresponds to a random network representation which is often non-trivial in many domains. However, we do see that in many domains the DAC-MDP performance further improves as the representation is learned, showing that the DAC-MDP framework is able to leverage improved representations.

Figures 4 investigates the performance at the final iteration for different values of $N_e$. For $N_e = 1$ we use $k = 5, k_\pi = 11, C = 1$ and for $N_e = 20$ we expand on the parameter set used for

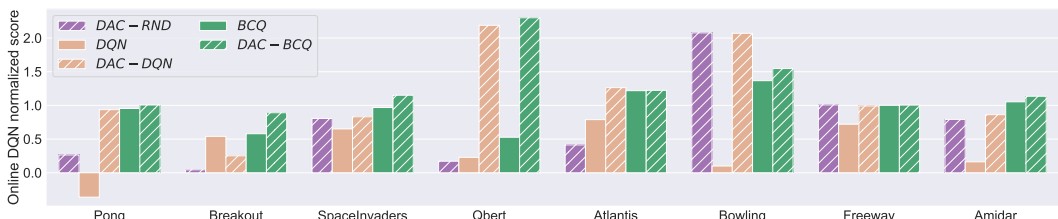

Figure 5: Atari results for 2.5M dataset. We show the final performance of BCQ and DQN trained for 2.5M iterations. We also use the same representation for the DAC-MDPs named as DAC-BCQ and DAC-DQN respectively. All DAC-MDPs are evaluated with $N_e = 6$.

$N_e = 6$ [$C \in \{1, 10, 100, 1000, 1M\}$, and $k_\pi \in \{5, 11, 31, 51\}$]. First we see that in the majority of cases, even $N_e = 1$ is able to perform as well or better than BCQ or DQN and that increasing $N_e$ often significantly improves performance. We do see that for three games, SpaceInvaders, QBert, and Atlantis, DAC-BCQ ($N_e = 1$) is significantly worse than the BCQ policy. This illustrates the potential challenge of hyperparameter selection in the offline framework without the benefit of additional online evaluations.

Finally we show results for all representation on the 2.5M dataset in Figure 5. In all but one case (Breakout DAC-DQN), we see that DAC-MDP improves performance or is no worse than the corresponding BCQ and DQN policies. Further in most cases, the DAC-MDP performance improves over the Online DQN agent used to generate the dataset which was trained on 10M online transitions. While random representations perform reasonably in some cases, they are significantly outperformed by learned representations. These results show, for the first time, that optimal planning can result in non-trivial performance improvement across a diverse set of Atari games. Given the simplicity of the representation learning and the difference between random and learned representations, these results suggest significant promise for further benefits from improved representations.

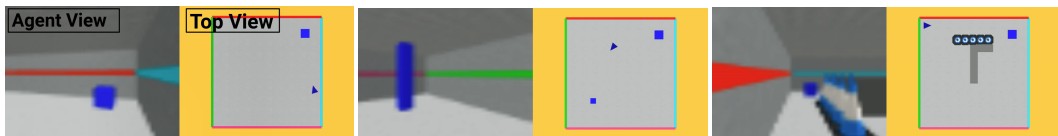

Figure 6: Agent and top view for 3D Navigation domains. (left) *Simple Room*, (center) *Box and Pillar Room* and (right) *Tunnel Room*.

**Illustrative Use Cases in 3D Navigation:** We now show the flexibility of an optimal planning approach using three use cases in Gym Mini-World (Chevalier-Boisvert, 2018), a first-person continuous 3D navigation environment. In all scenarios the agent has three actions: Move Forward, Turn Right, and Turn Left. Episodes lasted for a maximum of 100 time steps. The raw observations are 84x84 RGB images. We use random CNN representations in all experiments, which were effective for this environment. Our offline dataset was collected by following a random policy for 100K time steps. In these experiments we use $N_e = 2$ with $k = 5, k_\pi \in \{11, 51\}$, and $C = 0.01$. All the rooms used for the experiments and their optimal policies are visualized in fig 15a. We also compare our approach against baselines(DQN/BCQ) in Appendix A.6 and find that DAC-MDPs perform better and more consistently across the tasks.

*Case 1: Adapting to Modified Action Space.* We designed a simple room with a blue box in a fixed position and an agent that is spawned in a random initial position. The agent gets a -1 reward for bumping into the walls and a terminal reward of +1 for reaching the blue box. The DAC-MDP achieves an average total reward of 0.98 using the offline dataset. Next, we simulate the event that an actuator is damaged, so that the agent is unable to turn left. For most offline RL approaches this would require retraining the agent on a modified dataset from scratch. Rather our DAC-MDP approach can account for such a scenario by simply attaching a huge penalty to all left actions. The solution to this modified DAC-MDP achieves an average score of 0.96 and is substantially different than the original since it must find revolving paths with just the turn right action.

*Case 2: Varying Horizons.* We create a new room by adding a pillar in the simple room described above. Additionally, the agent gets a small non-terminal reward of 0.02 for bumping into the pillar.

Depending on the placement of the agent and the expected lifetime of the agent it may be better to go for the single +1 reward or to repeatedly get the 0.02 reward. This expected lifetime can be simulated via the discount factor, which will result in different agent behavior. Typical offline RL approaches would again need to be retrained for each discount factor of interest. Rather, the DAC-MDP approach simply allows for changing the discount factor used by VI and solving for the new policy in seconds. Using a small discount factor (0.95) our short-term agent achieves an average score of 0.91 by immediately attempting to get to the box. However, the long-term agent with larger discount factor (0.995) achieves a score of 1.5 by repeatedly collecting rewards from the pillar. Qualitative observation suggests that both policies perform close to optimally.

*Case 3: Robust/Safe Policies.* Our third room adds a narrow passage to the box with cones on the side. Tunnel Room simulates the classic dilemma of whether to choose a risky or safe path. The agent can risk getting a -10 reward by bumping into the cones or can optionally follow a longer path around an obstacle to reach the goal. Even with a relatively large dataset, small errors in the model can lead an agent along the risky path to occasionally hit a cone. To build robustness, we find an optimal solution to a modified DAC-MDP where at each step there is a 10% chance of taking a random action. This safe policy avoids the riskier paths where a random "slip" might lead to disaster and always chooses the safer path around the obstacle. This achieved an average score of 0.72. In contrast, the optimal policy for the original DAC-MDP achieved an average score of 0.42 and usually selected the riskier path to achieve the goal sooner (encouraged by the discount factor).

*Computation Time Analysis:* DAC-MDP has explicit convergence guarantees; however, the stopping condition of DQN and BCQ is not clear. We follow previous works and use batch updates equal to the size of the dataset. We define the first run for a set of hyperparameters/objective as an algorithm's *initial run*. We assume that the representation is learned in the initial run and frozen for secondary objectives. Here secondary objectives can include new hyperparameter sets or goals. Fig 7 compares the wall time of different approaches. Here we can see that DAC-MDPs are 2-4x faster for the initial run. For secondary objectives, however, we can see a wall-time

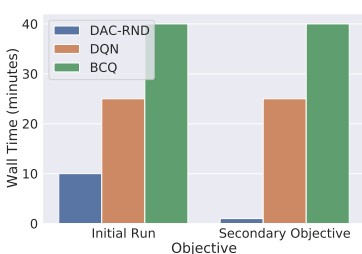

Figure 7: Comparison of Computation Cost for different appraoches. [100k Dataset]

speedup of anywhere between 25-40x. In the initial run of DAC-MDPs, KNN lookups[3] account for most of the computation time. Since we can reuse the KNN lookups for any secondary objective, the VI solver dominates the runtime for the new run, which is much faster. For a large dataset of 2.5M, it takes 60-90 minutes to compile the DAC-MDP and less than 3 minutes to solve the MDP.

The *memory consumption* scales linearly with the state count, action space, and the smoothness parameter k. Fully compiled MDPs for 2.5M approximately ranges from 2-6GB depending on the action space. The current implementation is not fully optimized for memory however one can perform state space aggregation in favor of memory efficiency.

## 5 RELATED WORK

**Averagers Framework.** Our work is inspired by the original averagers framework of Gordon (1995) which showed the equivalence of a derived MDP to a type of approximate dynamic programming. A construction similar to the derived MDP was explored in later work (Jong & Stone, 2007) within an online RL setting. A model similar to Gordon's derived MDP was later theoretically studied for efficient exploration in online continuous-state RL (Pazis & Parr, 2013). Due to its exploration goals, that work introduced bonuses rather than penalties for unknown parts of the space. Ormoneit & Glynn (2002) adapts fitted value iteration Gordon & Mitchell (1999) to kernel based reinforcement learning. Furthermore Ormoneit & Sen (2002) studied the theoretical convergence and consistency properties of this algorithm when combined with kernel-based regressors. Within the same framework Ernst et al. (2005) employed tree-based regression algorithms to show scalability in high-dimensional spaces. However, none of these approaches have been shown to scale to image based domains. Also, our work is the first to integrate the averagers derived MDP with pessimistic costs

---

[3]Currently, KNN lookups are performed by using CPU based KD-trees and has a throughput of 5̃k lookups/sec. However, leveraging GPU optimized KD-trees ((Martin, 2012), Hu et al. (2015)) seems promising to reduce the wall time of the algorithm further.

in order to make it applicable to offline RL. Our work is also the first to demonstrate the framework in combination with deep representations for scaling to complex observation spaces such as images.

**Optimal Planning with Deep Representations.** Recent work (Corneil et al., 2018) uses variational autoencoders with Gumbel softmax discretization to learn discrete latent representations. These are used to form a tabular MDP followed by optimal planning in an online RL setting. More recently, van der Pol et al. (2020) introduced a contrastive representation-learning and model-learning approach that is used to construct and solve a tabular MDP. Experimental results in both of these works were limited to a small number of domains and small datasets. Unlike these works, which primarily focus on representation learning, our focus here has been on providing a theoretically-grounded model, which can be combined with representation-learning. Works like Kurutach et al. (2018) are mostly focused on learning plannable representations than planning. Instead, they train a GAN to generate a series of waypoints which can then be is leveraged by simple feedback controllers. Also, Yang et al. (2020) and Agarwal et al. (2020) attempt to incorporate long term relationships into their learned latent space and employ search algorithms such as A* and eliptical planners.

**Episodic Control and Transition Graphs.** Several prior approaches for online RL, sometimes called episodic control, construct different types of explicit transition graphs from data that are used to drive model-free RL (Blundell et al., 2016; Hansen, 2017; Pritzel et al., 2017; Lin et al., 2018; Zhu et al., 2020; Marklund et al., 2020). None of these methods, however, use solutions produced by planning as the final policy, but rather as a way of improving or bootstrapping the value estimates of model-free RL. The memory structures produced by these methods have a rough similarity to deterministic DAC-MDPs with $k = 1$.

**Pessimism for Offline Model-Free Based RL.** Fonteneau et al. (2013) studied a "trajectory based" simulation model for offline policy evaluation. Similar in concept to our work, pessimistic costs were used based on transition distances to "piece together" disjoint observations, which allowed for theoretical lower-bounds on value estimates. That work, however, did not construct an MDP model that could be used for optimal planning. Most recently, in concurrent work to ours, pessimistic MDPs have been proposed for offline model-based RL (Kidambi et al., 2020; Yu et al., 2020). Both approaches define MDPs that penalizes for model uncertainty based on an assumed "uncertainty oracle" signal and derive performance bounds under the assumption of optimal planning. In practice, however, due to the difficulty of planning with learned deep-network models, the implementations rely on model-free RL, which introduces an extra level of approximation.

**Model free RL for Offline settings:** Agarwal et al. (2019) show that, for offline settings, stronger off policy methods stemming from distributional RL like REM and QR-DQN perform better to classic DQN. However recent offline RL methods directly tackle the problem of distributional shift in offline settings as pointed out by Atkeson (1998), and more recently Fonteneau et al. (2013). Works like Sutton et al. (2016),Nachum et al. (2019), Buckman et al. (2020) among others can directly estimate the policy gradient for offline learning. Moreover, Fujimoto et al. (2019), Wang et al. (2020), Kumar et al. (2020), Wu et al. (2019) directly constrains off policy learning towards the dataset, in turn, introducing pessimism in their estimates.

## 6 Summary

This work is an effort to push the integration of deep representation learning with near-optimal planners. To this end, we proposed the Deep Averagers with Costs MDP (DAC-MDP) for offline RL as a principled way to leverage optimal planners for tabular MDPs. The key idea is to define a non-parametric continuous-state MDP based on the data, whose solution can be represented as a solution to a tabular MDP derived from the data. This construction can be integrated with any deep representation learning approach and addresses model inaccuracy by adding costs for exploiting under-represented parts of the model. Using a planner based on a GPU-implementation of value iteration, we demonstrate scalability to complex image-based environments such as Atari with relatively simple representations derived from offline model-free learners. We also illustrate potential use-cases of our planning-based approach for zero-shot adaptation to changes in the environment and optimization objectives. Overall, the DAC-MDP framework has demonstrated the potential for principled integrations of planning and representation learning. There are many interesting directions to explore further, including integrating new representation-learning approaches and exploiting higher-order structure for the next level of scalability.

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

# A  APPENDIX

## A.1  PRACTICAL ISSUES

We now describe some of the key practical issues of the framework and how we address them.

**Optimal Planning.** We use value iteration (VI) (Bellman & Kneale, 1958) to optimally solve the core-state MDPs, which can have a number of states equal to the size of the dataset $N$. While in general storing a transition matrix with $N$ states can take $O(n^2)$ space, the core-state MDPs have sparse transitions with each state having at most $k$ successors, which our implementation exploits. Further, VI is highly parallelizable on GPUs, since the update of each state at each VI iteration can be done independently. We developed a simple GPU implementation of VI, which can provide between 20-1000x wall clock speedup over its serial counterpart. Our current implementation can solve MDPs with a million states in less than 30 seconds and easily scale to MDPs with several million states and medium action spaces (up to 10). Thus, this implementation is adequate for the dataset sizes that can be expected in many offline RL settings, where often we are interested in performing well given limited data. For extremely large datasets, we expect that future work can develop effective sub-sampling approaches, similar to those used for scaling other non-parametric models, e.g. Miche et al. (2009). We will release an open-source implementation and further details are in Appendix A.7.

**Weighted Averaging:** The above DAC-MDP construction and theory is based on uniform averaging across kNN sets. In practice, however, most non-parametric regression techniques use weighted averaging, where the influence of a neighbor decreases with its distance. Our implementation also uses weighted averaging, which we found has little impact on overall performance (see Appendix), but can reduce sensitivity to the choice of $k$, especially for smaller datasets. In our implementation we compute normalized weights over kNN sets according to the inverse of distances. In particular, the modified DAC-MDP reward and transition functions become,

$$\tilde{R}(s,a) = \sum_{i \in kNN(s,a)} \alpha(s,a,s_i,a_i)(r_i - C \cdot d(s,a,s_i,a_i)) \tag{4}$$

$$\tilde{T}(s,a,s') = \sum_{i \in kNN(s,a)} \alpha(s,a,s_i,a_i)\ I[s' = s_i'], \tag{5}$$

where $\alpha(s,a,s_i,a_i) = \frac{d'(s,a,s_i,a_i)}{\sum_{j \in kNN(s,a)} d'(s,a,s_j,a_j)}$, $d'(s,a,s_i,a_i) = \frac{1}{d(s,a,s_i,a_i)+\delta_d}$ and $\delta_d = 1e^{-5}$

**Optimized Policy Computation.** After solving the DAC-MDP, the policy computation for new states requires computing Q-values for each action via Equation 3, which involves a kNN query for each action. We find that we can reduce this computation by a factor of $|A|$ via the following heuristic that performs just one kNN query at the state level. Here $kNN(s)$ returns the data tuple of indices whose source states are the $k$ nearest neighbors of $s$.

$$\tilde{\pi}(s) = \max_{a \in A} \frac{1}{k} \sum_{i \in kNN(s)} \alpha(s,s_i)\ \tilde{Q}^*(s_i',a) \tag{6}$$

where, $d'(s,s_i) = \frac{1}{d(s,s_i)+\delta_d}$, $\alpha(s,s_i) = \frac{d'(s,s_i)}{\sum_{i \in kNN(s)} d'(s,s_i)}$ and $\delta_d$ is set to $1e^{-5}$. We have found that this approach rarely hurts performance and results in the same action choices. However, when there is limited data, there is some evidence that this computation can actually help since it avoids splitting the dataset across actions (see the Appendix for ablation study).

**Hyperparameter Selection.** The choice of the cost parameter $C$ can significantly influence the DAC-MDP performance. At one extreme, if $C$ is large, then the resulting policy will focus on "staying close" to the dataset. At the other extreme when $C = 0$, there is no penalty for exploiting the under-represented parts of the model. In general, the best choice will lie between the extremes and must be heuristically and/or empirically selected. Here, We use the simple rule-of-thumb of setting $C$ to be in the order of magnitude of the observed rewards, to avoid exploitation but not dominate the policy. In addition, if the online evaluation budget $N_e$ (see Section 2) is greater than one, we also consider values that are orders of magnitude apart to span qualitatively different ranges.

The DAC-MDP construction used the same smoothness parameter $k$ for both building the MDP and computing the policy for unknown states via Equation 3. We have found it is useful to use different

values and in particular, there can be a benefit for using a larger value for the later use to reduce variance. Thus, our experiments will specify a value of $k$ used to construct the MDP and a value $k_\pi$ used to compute the policy. Our default setting is $k = 5$ and $k_\pi = 11$, with more values possibly considered depending on the evaluation budget $N_e$.

## A.2 THEORETICAL PROOFS

Let $\tilde{\pi}$ bet the optimal policy of a DAC-MDP. We wish to bound the value $V^{\tilde{\pi}}(s)$ of $\tilde{\pi}$ in the true MDP in terms of the optimal value $V^*(s)$ for any state $s$. Using the following lemma from Pazis & Parr (2013) it is sufficient to bound the *Bellman Error* $\tilde{Q}(s, a) - B[\tilde{Q}](s, a)$ of $\tilde{Q}$ across all $s$ and $a$ with respect to the true MDP.

**Lemma A.1.** *(Pazis & Parr, 2013),Theorem 3.12 For any Q-function Q with greedy policy $\pi$, if for all $s \in \mathcal{S}$ and $a \in \mathcal{A}$, $-\epsilon_- \leq Q(s, a) - B[Q](s, a) \leq \epsilon_+$, then for all $s \in \mathcal{S}$,*

$$V^\pi(s) \geq V^*(s) - \frac{\epsilon_- + \epsilon_+}{1 - \gamma}$$

.

In general, however, the Bellman Error can be arbitrarily large without further assumptions. Thus, in this work, we make Lipschitz smoothness assumptions on $B[\tilde{Q}]$. In particular, we assume that there is a constant $L(k, C)$ such that for any state-action pairs $(s, a)$ and $(s', a')$ we have

$$\left| B[\tilde{Q}](s, a) - B[\tilde{Q}](s', a') \right| \leq L(k, C) \cdot d(s, a, s', a').$$

Given the smoothness assumption for any $(s, a)$, data sample $(s_i, a_i, r_i, s'_i)$, and a constant $L$ we define a constant $\Delta_i(s, a)$ such that

$$B[\tilde{Q}](s, a) = B[\tilde{Q}](s_i, a_i) - \Delta_i(s, a).$$

Note that based on the smoothness assumption we have that $|\Delta_i(s, a)| \leq L(k, C) \cdot d(s, a, s_i, a_i)$.

Using this definition we introduce a new operator $\hat{B}$, which will allow us to relate $\tilde{B}$ to $B$.

$$\hat{B}[Q](s, a) = \frac{1}{k} \sum_{i \in kNN(s,a)} r_i + \gamma \max_{a'} Q(s'_i, a') - \Delta_i(s, a). \tag{7}$$

The following lemma shows that this operator approximates $B$ with high probability.

**Lemma A.2.** *For any data set $\mathcal{D}$ of size $N$, value of $k$, and any cost parameter $C$, if $\tilde{Q}$ is the optimal Q-function of the corresponding DAC MDP, then with probability at least $1 - \delta$, for all $(s, a) \in \mathcal{S} \times \mathcal{A}$,*

$$\hat{B}[\tilde{Q}](s, a) - B[\tilde{Q}](s, a) \leq Q_{max}\epsilon(k, N, \delta) \text{ for all } (s, a) \in \mathcal{S} \times \mathcal{A}$$

$$\epsilon(k, N, \delta) = \sqrt{\frac{1}{2k} \ln \frac{2M_{N,k}}{\delta}}$$

*where $Q_{max} = \max_{(s,a)} \tilde{Q}(s, a)$ and $n$ is the dimensionality of the state-action encoding.*

*Proof.* In order to capture the variance of $\hat{B}[\tilde{Q}]$ associated with the transition dynamics, for each state-action pair $(s, a)$ and each nearest neighbor index $i \in kNN(s, a)$, define a random variable $X_i(s, a) = r_i + \gamma \max_{a'} \tilde{Q}(S'_i, a') - \Delta_i(s, a)$, where $S'_i \sim T(s_i, a_i, \cdot)$. Note that each term of $\hat{B}[\tilde{Q}](s, a)$ is a single sample of one $X_i(s, a)$. That is,

$$\hat{B}[\tilde{Q}](s, a) = \frac{1}{k} \sum_{i \in kNN(s,a)} x_i(s, a), \text{ where } x_i(s, a) \sim X_i(s, a).$$

Also note that according to the definition of $\Delta_i(s,a)$ the expected value of each $X_i(s,a)$ is equal to $B[\tilde{Q}](s,a)$.

$$\mathbb{E}\left[X_i(s,a)\right] = r_i + \gamma\mathbb{E}\left[\max_{a'} \tilde{Q}(S_i', a')\right] - \Delta_i(s,a)$$
$$= B[\tilde{Q}](s_i, a_i) - \Delta_i(s,a)$$
$$= B[\tilde{Q}](s,a)$$

Accordingly $\mathbb{E}\left[\hat{B}[\tilde{Q}](s,a)\right] = B[\tilde{Q}](s,a)$.

From the above, we can apply the Hoeffding inequality, which states that for $k$ independent random variables $X_1, \ldots, X_k$ with bounded support $a_i \leq X_i \leq b_i$, if $\bar{X} = \frac{1}{k}\sum_i X_i$ is the empirical mean, then for all $\epsilon > 0$,

$$Pr\left(\left|\bar{X} - \mathbb{E}\left[\bar{X}\right]\right| \geq \epsilon\right) \leq 2\exp\left(\frac{-2k^2\epsilon^2}{\sum_i (b_i - a_i)^2}\right).$$

Applying this bound to $\hat{B}[\tilde{Q}](s,a)$ implies that:

$$Pr\left(\left|\hat{B}[\tilde{Q}](s,a) - B[\tilde{Q}](s,a)\right| \geq \epsilon\right) \leq 2\exp\left(\frac{-2k\epsilon^2}{Q_{max}^2}\right),$$

which can be equivalently written as

$$Pr\left(\left|\hat{B}[\tilde{Q}](s,a) - B[\tilde{Q}](s,a)\right| \geq Q_{max}\sqrt{\frac{1}{2k}\ln\frac{2}{\delta'}}\right) \leq \delta'.$$

This bound holds for individual $s,a$ pairs. However, we need to bound the probability across all $s,a$ pairs. To do this note that the computed value $\hat{B}[\tilde{Q}](s,a)$ is based on the nearest neighbor set $kNN(s,a)$ and let $M_{N,k}$ denote an upper bound on the possible number of those sets across all $\mathcal{S} \times \mathcal{A}$. To ensure that the bound holds simultaneously for all such sets, we can apply the union bound using $\delta' = \delta/M_{N,k}$. This bounds the probability over all state-action pairs simultaneously by $\delta$.

$$Pr\left(\left|\hat{B}[\tilde{Q}](s,a) - B[\tilde{Q}](s,a)\right| \geq Q_{max}\sqrt{\frac{1}{2k}\ln\frac{2M_{N,k}}{\delta}}\right) \leq \delta.$$

$\square$

**Theorem 3.1** For any data set $\mathcal{D}$ of size $N$, let $\tilde{Q}$ and $\tilde{\pi}$ be the optimal Q-function and policy for the corresponding DAC-MDP with parameters $k$ and $C$. If $B[\tilde{Q}]$ is Lipschitz continuous with constant $L(k,C)$, then with probability at least $1 - \delta$,

$$V^{\tilde{\pi}} \geq V^* - \frac{2\left(L(k,C) \cdot \bar{d}_{max} + Q_{max}\epsilon(k,N,\delta)\right)}{1-\gamma},$$

$$\epsilon(k,N,\delta) = \sqrt{\frac{1}{2k}\ln\frac{2M_{N,k}}{\delta}},$$

which for L2 distance over a $d$-dimensional space yields $\epsilon(k,N,\delta) = O\left(\sqrt{\frac{1}{k}\left(d\ln kN + \ln\frac{1}{\delta}\right)}\right)$.

*Proof.* The proof first will bound the Bellman error of $\tilde{Q}$ from above an below and then apply Lemma A.1. We first decompose the Bellman error into two parts by adding and subtracting $\hat{B}[\tilde{Q}]$.

$$\tilde{Q}(s,a) - B[\tilde{Q}](s,a) = \underbrace{\left(\tilde{Q}(s,a) - \hat{B}[\tilde{Q}](s,a)\right)}_{\xi_d(s,a)} + \underbrace{\left(\hat{B}[\tilde{Q}](s,a) - B[\tilde{Q}](s,a)\right)}_{\xi_{sim}(s,a)}$$

The first term corresponds to the error due to the non-zero distance of the $k$ nearest neighbors and the second term is due to sampling error.

Noting that $\tilde{B}$ and $\hat{B}$ only differ in one term, $\xi_d(s,a)$ can be simplified as follows.

$$\xi_d(s,a) = \tilde{Q}(s,a) - \hat{B}[\tilde{Q}](s,a)$$
$$= \tilde{B}[\tilde{Q}](s,a) - \hat{B}[\tilde{Q}](s,a)$$
$$= \frac{1}{k} \sum_{i \in kNN(s,a)} \Delta_i(s,a) - C \cdot d(s,a,s_i,a_i)$$

From this and the fact tht $|\Delta_i(s,a)| \le L(k,C) \cdot d(s,a,s_i,a_i)$ we can immediately derive upper and lower bounds on $\xi_d(s,a)$ for all $s$ and $a$.

$$-(L(k,C)+C) \cdot \bar{d}_{max} \le \xi_d(s,a) \le (L(k,C)-C) \cdot \bar{d}_{max}$$

We can bound $\xi_{sim}(s,a)$ by a direct application of Lemma A.2. Specifically, with probability at least $1-\delta$, for all $s$ and $a$, $|\xi_{sim}(s,a)| \le \epsilon(k,N,\delta)$. Putting these together we get that with probability at least $1-\delta$, for all $s$ and $a$,

$$-\big((L(k,C)+C) \cdot \bar{d}_{max} + \epsilon(k,N,\delta)\big) \le \tilde{Q}(s,a) - B[\tilde{Q}](s,a) \le (L(k,C)-C) \cdot \bar{d}_{max} + \epsilon(k,N,\delta).$$

The proof is completed by applying Lemma A.1 to this bound.

Note: For a Euclidean distance metric, Toth et al. (2017)[Chapter 27] has established an upper bound on $M_{N,k}$.

$$M_{N,k} = O\left(N^{\lceil d/2 \rceil} k^{\lfloor d/2 \rfloor + 1}\right) = O\left((kN)^{d/2+1}\right).$$

$\square$

### A.3 ABLATION STUDY

We highlight the deviation of the practical implementation of DAC-MDPs from theory in section A.1, namely, weighted averaging based on KNN distances ($WA$) and, KNN over states instead of state-action pairs ($sKNN$). The theory for DAC-MDP is derived for uniform averaging. i.e., the probability distribution to the candidate next states from KNN state action pairs is uniformly distributed irrespective of their relative distances. In contrast, weighted averaging normalizes the probability distribution according to their relative distances from the query state-action pair. Secondly, the theory states that we query the K nearest neighbors for each state-action pair to calculate the q values for any unseen state. This entails that $|A|$ numbers of KNN calls have to be made for each action decision. However, we can reduce this by simply querying k nearest neighbors for states over state-action pairs. We query for K- nearest states when $sKNN$ option is turned on. We conduct the ablation study on the full[100k] dataset as well as a smaller dataset comprising of only 10% of the full dataset. Below in Figure 8 we show ablation study for each of these choices in the standard CartPole domain.

We find that for a larger dataset, neither weighted averaging nor the state KNN approximation affects the performance of DAC-MDPs. However, there is a noticeable drop in performance for $optimalBag$ dataset for smaller dataset sizes when state KNN approximation is turned off. This suggests that when the dataset is mostly comprised of subsamples of optimal trajectories, the data is not divided uniformly over the actions, resulting in less accurate estimates, especially when the data is limited and skewed towards a particular policy.

### A.4 ADDITIONAL CARTPOLE EXPERIMENTS

To further investigate the impact of DAC-MDP parameters, we conduct similar experiments to section 4 for different dataset sizes on CartPole. In addition to the greedy policy performance we also track the performance of $\epsilon$-greedy run of the policy to further distinguish the quality/robustness of the policies learned.

We see a very similar trend (as in 100k) for the choice of cost parameter $C$ even for the low data regime of 10k. Figure 9 shows that for all datasets, when there is no penalty for under-represented transitions,i.e. $C = 0$, the policy is extremely poor. At the other extreme, when $C$ is very large, the optimal policy tries to stay as close as possible to transitions in the dataset. This results in good performance for the $OptimalBag$ dataset, since all actions are near optimal. However, the policies

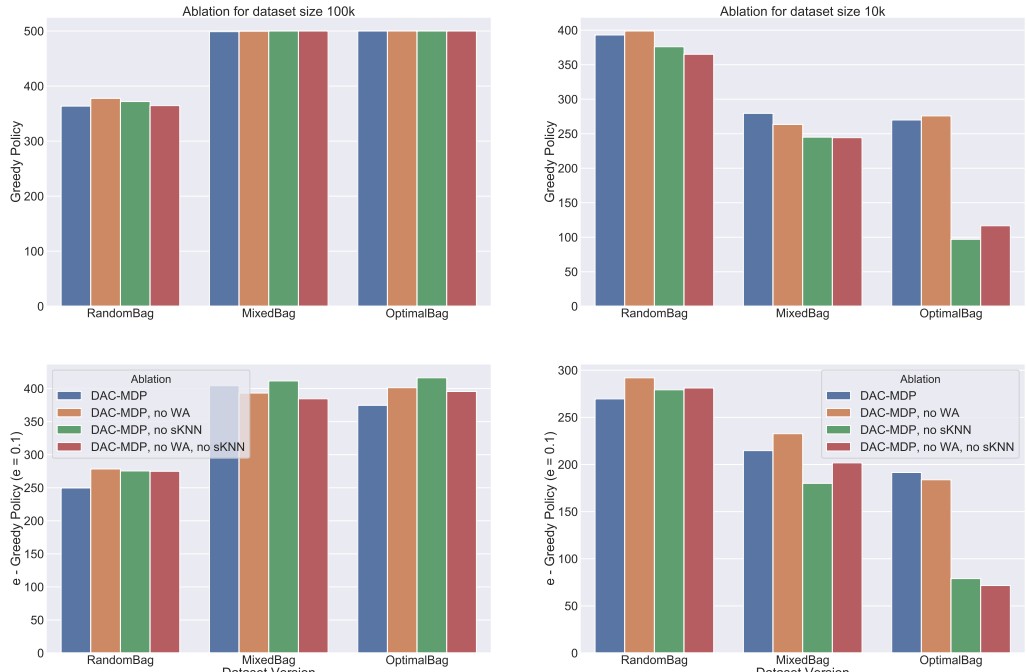

Figure 8: (a) Ablation study for WA and sKNN in CartPole Domain. Greedy and eps-greedy policy returns for different sets of hyperparameters and dataset versions of size 100k. (left) and 10K (right) Hyperparameters: $[k = 5, k_\pi = 11, C = 1, N_e = 1]$

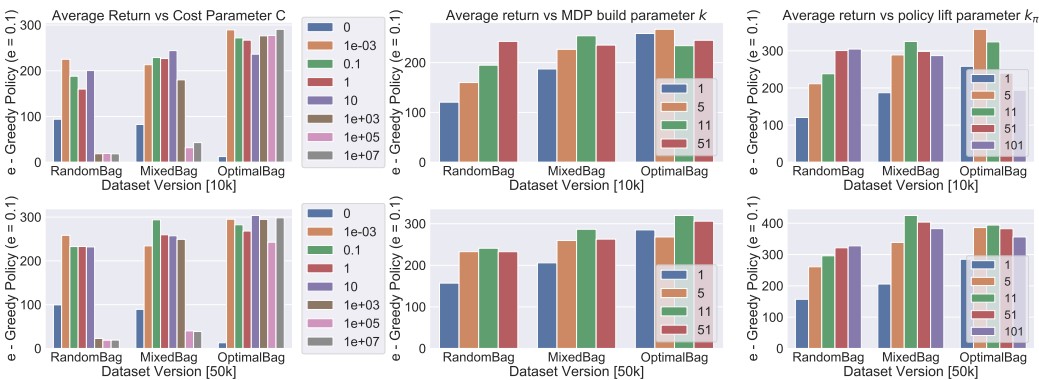

Figure 9: Eps-Greedy performance for CartPole on different dataset sizes. (a) Top row: dataset size 10k (b) Bottom Row: dataset size 50k

resulting from the $MixedBag$ and $RandomBag$ datasets fail. This is due to those datasets containing a large number of sub-optimal actions, which the policy should actually avoid for purposes of maximizing reward. Between these extremes, the performance is relatively insensitive to the choice of $C$.

DAC-MDP, however, is more sensitive towards the choice of k for small dataset regime. Figure 9 second column, explores the impact of varying k using fixed $k_\pi = 1$ and $C = 1$. The main observation is that there is a slight disadvantage to using $k = 1$, which defines a deterministic finite MDP, compared to $k > 1$, especially for the non-optimal datasets. Moreover, there is also a slight disadvantage in using larger k when the dataset is not dense. It is to be noted that although $RandomBag$ dataset contains the same amount of experience, it is much denser than other datasets as random trajectories are quite short compared to optimal trajectories. This indicates that the optimal planner

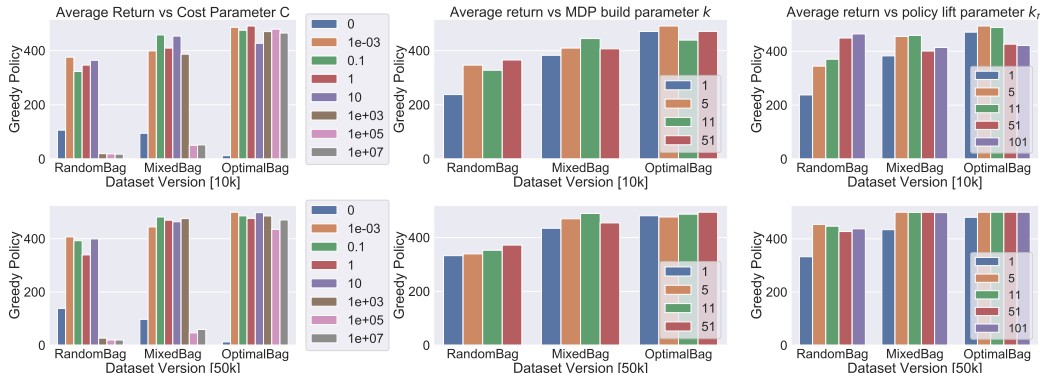

Figure 10: Greedy performance for CartPole on different dataset sizes. (a) Top row: datset size 10k (b) Bottom Row: dataset size 50k

is able to benefit by reasoning about the stochastic transitions of the DAC-MDP for $k > 1$. However, DAC-MDP suffers from high values of k when dataset is sparse.

Finally Figure 9 third column, varies the policy smoothing parameter $k_\pi$ from 1 to 101. Similar to the results for varying smoothness parameter k, there is a benefit to using a value of $k_\pi > 1$, but should not be chosen to be too large depending on how large the dataset is.

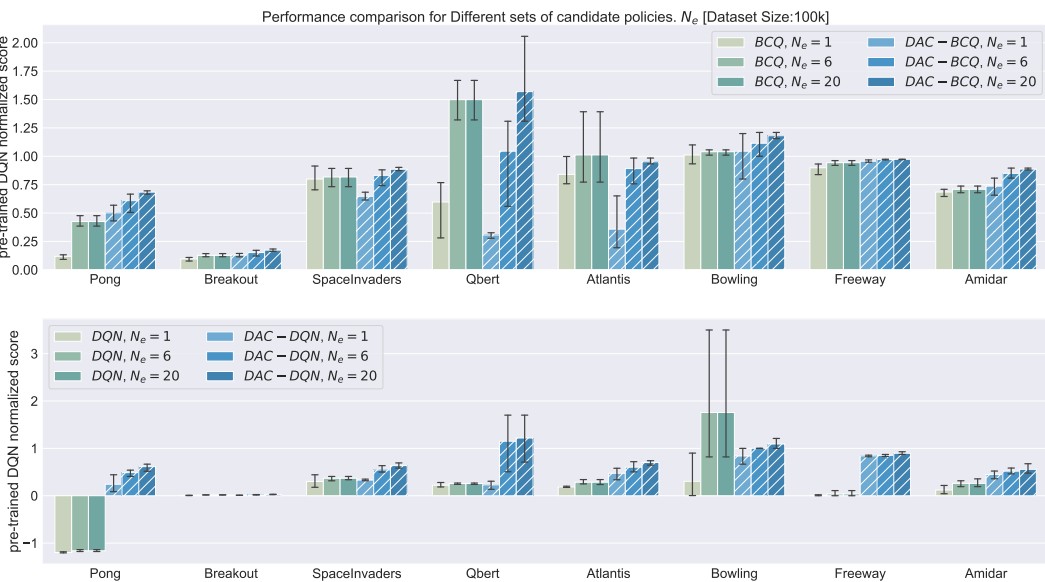

Figure 11: (a) Greedy performance for Atari using different learnt representations and evaluation candidate policies $N_e$[100k dataset]. Runs averaged over 3 runs. Error bars show the 95% confidence interval.

## A.5  ADDITIONAL ATARI EXPERIMENTS

**Policy search for DQN and BCQ**: In our experiments we consider an evaluation protocol that makes the amount of online access to the environment explicit. In particular, the offline RL algorithm is allowed to use the environment to evaluate $N_e$ policies (e.g., each evaluation can be an average over repeated trials), which, for example, may derived from different hyperparameter choices. It is not clear how many of these evaluations will be actually needed for Q-learning algorithms such as DQN that is primarily designed for online learning. Even approaches focusing on offline learning are

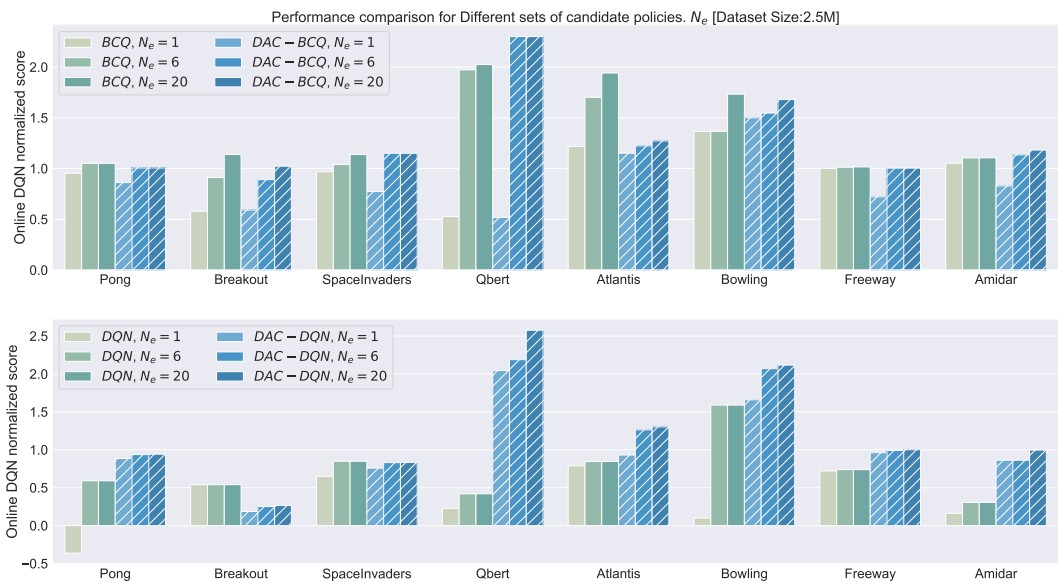

Figure 12: (a) Greedy performance for Atari using different learnt representations and evaluation candidate policies $N_e$ [2.5M dataset]

not spared from hyperparameter search and stopping criteria. Hence it is not clear how to evaluate Q-iterating policies such as DQN or BCQ for different values of $N_e$ where we are already using the best parameters reported from previous works.

However, we can still assume that we know the value of $N_e$ before hand and tune the learning process accordingly. More specifically given a fixed number of training iterations the evaluation frequency can be set such that we complete $N_e$ evaluations by the end of the training. We can then use the last $N_e$ evaluation checkpoints to obtain the set of candidate policies. We can then choose the best among them. It is worth noting that, even if we disregard the hyperparameter tuning, it is still not entirely fair to compare BCQ,$N_e = 6$ directly with DAC-BCQ, $N_e$=6 as the DAC-MDP only has access to the very last representation. Moreover, DAC-MDPs do not need stopping criteria and are more robust to small representational changes. We show the policy search results for both DQN and BCQ for 100k dataset as well as 2.5M dataset in Figure 11 and Figure 12 respectively.

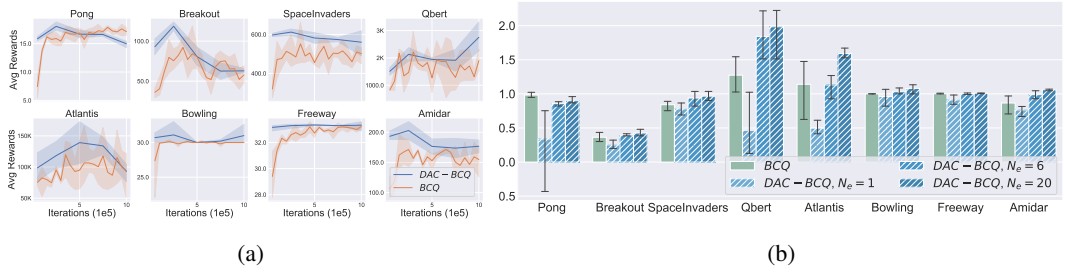

Figure 13: Results on Medium dataset size of 1M for (a) BCQ representation and BCQ agent is trained for 1m timesteps, and evaluated on 10 episodes every 50k steps. At each of 6 uniformly distributed evaluation checkpoints, we use the internal representation to compile DAC-MDPs. We then evaluate the DAC-MDPs for $N_e = 6$. (b) Final BCQ policy along with the corresponding performance of DAC-BCQ for different values of $N_e$. Runs averaged over 3 runs, error bars so the 95% confidence interval.

Additionally we also run the experiment on dataset size 1M at different evaluation checkpoints as done in the main paper. We trained offline $BCQ$ agent on the dataset for 1M iterations. This allows us to compare to these baselines as well as use the $BCQ$ latent state representations for

DAC-MDP construction. The performance of the corresponding $BCQ$ policies is evaluated using 10 test episodes every 50K iterations. At some select position of $BCQ$ evaluation iteration, we constructed DAC-MDPs using the latent state representation at that point. In particular, this first experiment considers the offline RL setting where $N_e = 6$, meaning that we can evaluate 6 policies(using 10 episodes each) corresponding to 6 different DAC-MDP parameter settings comprised of the combinations of $k = 5, C \in 1, 100, 1M$, and$k_\pi \in 11, 51$. For each representation the best performing DAC-MDP setting at each point is then recorded. The entire 1M iteration protocol was repeated 3 times and Figure 13a show the averaged curves with 90%confidence intervals. Figure 13b investigates the performance of final iteration for different values of $N_e$. All hyperparameters and normalization were selected as in the 100K experiment. We see that in the DAC-MDPs can perform as good as BCQ or better in most of the evaluation checkpoints even for the larger dataset size and training iterations.

## A.6 ADDITIONAL 3D NAVIGATION EXPERIMENTS

We also compare DAC-MDPs performance against DQN and BCQ baselines for first person 3D navigation Domain. In all scenarios the agent has three actions: Move Forward,Turn Right, and Turn Left and a maximum episode time steps of 100. The raw observations are 84x84 RGB images. We use the same settings for DAC-MDP as described in the main paper. We use the same baselines DQN and BCQ from Atari experiments with different set of hyperparameters as detailed in Appendix B.2

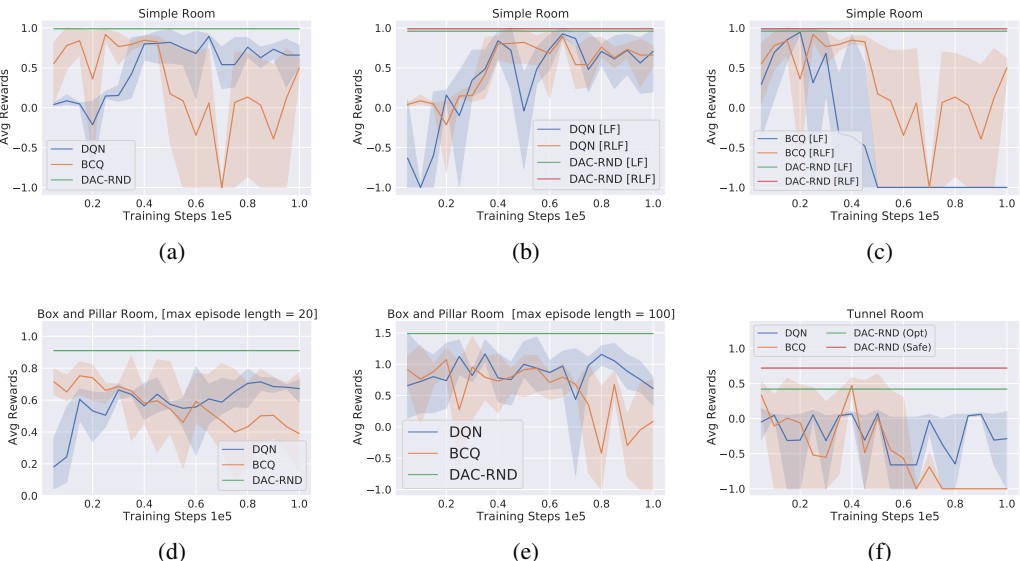

Figure 14: Comparison of DAC-MDPs using randomly initialized representation networks (DAC-RND) with baselines DQN and BCQ for 3D navigation Domains. (a) Simple Room Domain, (b) Simple Room Domain: DQN vs DAC-RND on right actuator failure[LF] (c) BCQ vs DAC-RND on right actuator failure[LF](d) Box and Pillar Room: short horizon, maximum episode length of 20, (e) Box and Pillar Room: Long horizon, maximum episode lengths of 100, (f) Tunnel Room. Rewards across the rooms are clipped at (-1,2) for normalized results.

*Case 1: Adapting to Modified Action Space.* Here an agent is spawned in a random initial position with a goal(blue box) in a fixed position. We call this Simple Room $(V^{\pi_\beta} = -9.36)$[4]. The agent gets a -1 reward for bumping into the walls and a terminal reward of +1 for reaching the goal. We run DQN and BCQ on the same dataset in the simple room. figure this shows that DAC-MDP performs better. For the simulated event of damaged actuator, we simply discard the damaged action from the policy action space. While this allows for a zero-shot transfer learning in this scenario, the methods

---

[4]$V^{\pi_\beta}$ is the expected performance of the behavioral policy for the dataset.

suffer with higher variance for DQN and fails to perform well in case of BCQ as pointed in figure this and this. The DAC-MDP however can adapt to these changes very robustly.

*Case 2: Varying Horizons.* We create a new room by adding a pillar in the simple room described above. Additionally, the agent gets a small non-terminal reward of 0.02 for bumping into the pillar. We call this BoxAndPillar Room ($V^{\pi_\beta} = -9.84$). Depending on the placement of the agent and the expected lifetime of the agent it may be better to go for the single +1 reward or to repeatedly get the 0.02 reward. This expected lifetime can be simulated via the discount factor, which will result in different agent behavior. As the baselines do not allow for zero shot transfer in this setting, we retrain the DQN and BCQ agents for different discount factors: 0.95 for short horizon and 0.99 for long horizon task. We then evaluate these policies on BoxAndPillar room with different maximum episode steps. 20 and 100 for short and long horizon settings respectively.

Here as pointed in figure this and this we see that the baselines cannot account for the long horizon planning effectively and mostly choose to plan for the shorter horizon even with a larger discount factor. hence it is clear that the discount factor do not translate to effective planning horizon in these baselines. However, the DAC-MDP approach using a small discount factor (0.95) our short-term agent achieves an average score of 0.91 learns to immediately get to the box along with the long-term agent repeatedly collecting rewards from the pillar. Moreover DAC-MDP simply allows for changing the discount factor used by VI and solving for the new policy in seconds for this secondary objective.

*Case 3: Robust/Safe Policies.* The Tunnel room ($V^{\pi_\beta} = -25$) adds a narrow passage to the box with cones on the side. This room simulates the classic dilemma of whether to choose a risky or safe path. The agent can risk getting a -10 reward by bumping into the cones or can optionally follow a longer path around an obstacle to reach the goal. Even with a relatively large dataset, small errors in the model can lead an agent along the risky path to occasionally hit a cone. We train a DQN and BCQ agent using the same dataset and see that they do not perform as well as DAC-DQN in the experiments. Moreover, it is not straight forward on how to add stochasticity in the baselines to achieve a more robust policy. In contrast, we can easily find an optimal solution to a modified DAC-MDP where at each step there is a 10% chance of taking a random action. This safe policy avoids the riskier paths where a random "slip" might lead to disaster and always chooses the safer path around the obstacle, and also achieves an average score of 0.72.

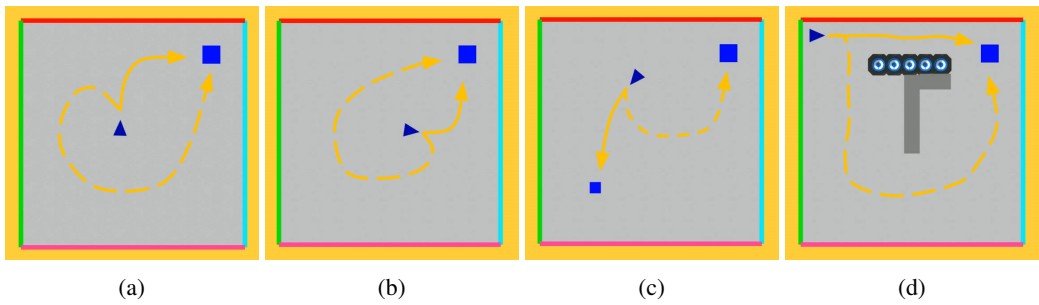

(a)  (b)  (c)  (d)

Figure 15: Visualization of DAC-MDP policies for 3D Navigation Domain (a) **Simple Room** *Solid arrow*: policy rollout of standard DAC-MDP. *Dotted arrow:* policy rollout of modified DAC-MDP ;Right Turn Penalized . (b) **Simple Room** *Solid arrow:* policy rollout of standard DAC-MDP. *Dotted arrow:* policy rollout of modified DAC-MDP; Left Turn Penalized . (c) **Box and Pillar Room** *Dotted Arrow:* policy rollout of DAC-MDP solved with small discount factor [short-term planning]. *Solid Arrow:* Policy rollout of DAC-MDP solved with Large Discount Factor [long-term planning] (d) **Tunnel Room** *Dotted Arrow:* Policy rollout of standard DAC-MDP. No stochasticity Introduced. [optimal policy]. *Solid Arrow:* Policy rollout of modified DAC-MDP with added stochasticity in dynamics. [safe policy]

Hence we see that typical Deep RL methods like DQN and BCQ do not excel for secondary objectives. There are settings such as failed actuator where they can perform zero-shot transfer learning but fail to give any guarantees, moreover they can fail sometimes as shown empirically. In cases where zero-shot transfer is not possible such as different horizons, they can be retrained for these secondary objectives, but this comes at a huge cost (if and when they succeed) as compared to just solving for a new objective in DAC-MDP as shown in the computational analysis in the main paper.

Furthermore, these baseline methods are not able to tackle objectives such as robustness in any trivial manner. In contrast DAC-MDP approaches have a fine-tuned control over the learned MDP and its planning parameters.

## A.7 SCALING VALUE ITERATION WITH GPUS

Value iteration (VI) (Bellman & Kneale, 1958) successively approximates the value function, starting from an arbitrary initial estimate by turning the Bellman optimality equation into an update rule. While VI is simple and able to calculate the exact solution for any MDP, one of the major disadvantages is that they are computationally inefficient. A large number of bellman backups has to be computed for each state before convergence and the memory required grows exponentially with the number of states in the MDPs. Hence, it is only natural to optimize value iteration using parallel computing platforms such as GPUs. ór Jóhannsson (2009) was one of the first to introduce a GPU based MDP solver. In particular, they introduce two algorithms: Block Divided Iteration and Result Divided Iteration. However, there have been works leveraging GPU optimized value iteration for specific domains. (eg. Ruiz & Hernández (2015), Wu et al. (2016)), there does not exist a standard public library for GPU optimized value iteration. To facilitate this, we implement a version that is a hybrid between the Block Divided Iteration and Result Divided Iteration approaches using a simple CUDA kernel as described in pseudo-code 1[5].

---

**Pseudocode 1** GPU Value Iteration Kernel

1: **procedure** BELLMANBACKUP($*T_P, *T_I, *R, *V, *V', *Q', *\delta$)
2:     $i \leftarrow get\_thread\_id()$
3:     $v, v_{max} \leftarrow 0, 0$
4:     **for** $j \in range(A)$ **do**
5:         **for** $k \in range(k_b)$ **do**   ▷ $k_b$ is initialized externally as per the MDP build parameter $k_b$
6:            $P_{ss'} \leftarrow T_P[i, j, k]$
7:            $I_{s'} \leftarrow T_I[i, j, k]$
8:            $v \leftarrow v + P_{ss'}V[I_{s'}]$
9:         **end for**
10:         $Q'[i, j] \leftarrow v$
11:         $v_{max} \leftarrow max(v, v_{max})$
12:     **end for**
13:     $V'[i] \leftarrow v_{max}$
14:     $\delta[i] = abs(V[i] - V'[i])$
15: **end procedure**

---

**Pseudocode 2** GPU Value Iteration Function

1: **procedure** VALUEITERATION($tranDict, rewardDict, \delta_{min}$)
2:     $T_p, T_I, R = get\_sparse\_representation(tranDict, rewardDict)$
3:     $T_p, T_I, R = allocate\_gpu\_memory(T_p, T_I, R)$
4:     $V, Q', \delta = allocate\_gpu\_memory\_for\_value\_vectors()$
5:     **while** $min(\delta) > \delta_{min}$ **do**
6:         $V' = allocate\_gpu\_memory(V.size())$
7:         $RunGPUKernel(T_p, T_I, R, V, V', Q', \delta)$
8:         $V = V')$
9:         $\delta[i] = abs(V[i] - V'[i])$
10:         $release\_memory\_for(V')$
11:     **end while**
12: **end procedure**

---

For a compact representation of the Transition matrix, we use a list of lists as our sparse representations. Here the Transition Matrix is divided into two matrices, one that holds the index of next states with a non-zero transition probability ($T_I$) and the other, which holds the actual transition

---

[5]Note that the different versions of GPU implementation does not affect the convergence properties of Value Iteration as each approach is still carrying out exact value iteration.

probability($T_P$). Each thread takes a single row from these two matrices and the Reward matrix to compute each state-action pair's Q values and the new value of a state. The value vector is shared among the threads and synchronized after each bellman backup operation.

To benchmark the GPU implementation's performance, we compare its run-time with a standard serial implementation of Value iteration across MDPs of varying sizes. These MDPs are DAC-MDPs generated by different samples from a large pool of datasets with continuous state vectors. The serial implementation is run on an Intel Xeon processor and does not use any CPU multi-processing. We plot the relative performance gain across different GPUs with varying CUDA cores. We consider 3 GPUs, namely, GTX 1080ti, RTX 8000, and Tesla V100, each with a CUDA core count of 3584, 4608 and 6912, respectively. The GPU optimized implementation provides anywhere between 20-1000X boost in solving speed over its serial counterpart, as shown in Figure 16. Currently, our implementation of VI can solve MDPs with a million states less than 30 seconds.

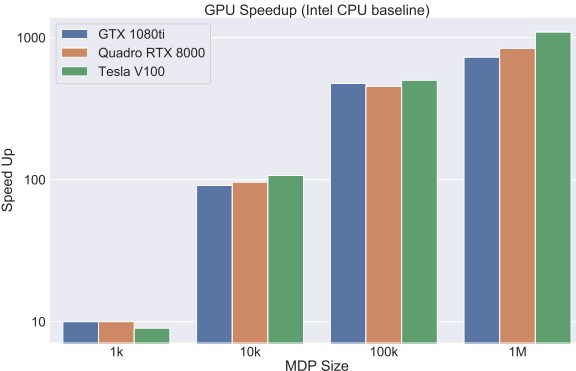

Figure 16: Compares the performance results for serial VI solvers with its GPU optimized implementation. The serial VI solver is benchmarked using Intel Xeon CPU. We plot the performance gain over different MDP sizes.

## B  EXPERIMENTAL DETAILS

### B.1  ATARI PREPROCESSING

The Atari 2600 environment is processed in teh same manner as previous work (Mnih et al., 2015; Machado et al., 2018) and we use consistent preprocesssing across all tasks and algorithms.

Table 1: Atari pre-processing details

| Name | Value |
|---|---|
| Sticky actions | Yes |
| Sticky action probability | 0.25 |
| Grey-scaling | True |
| Observation down-sampling | (84,84) |
| Frames stacked | 4 |
| Frameskip (Action repetitions) | 4 |
| Reward clipping | [-1,1] |
| Terminal condition | Game Over |
| Max frames per episode | 108K |

### B.2  ARCHITECTURE AND HYPER-PARAMETERS

Same architecture and hyperparameters were used as in Fujimoto et al. (2019) with slight modifications in the architecture.

Table 2: Architecture used by each Network

| Layer | Number of outputs | Other details |
|---|---|---|
| Input frame size | (4x84x84) | — |
| Downscale convolution 1 | 12800 | kernel 8x8, depth 32, stride 4x4 |
| Downscale convolution 2 | 5184 | kernel 4x4, depth 32, stride 2x2 |
| Downscale convolution 3 | 3136 | kernel 3x3, depth 32, stride 1x1 |
| Hidden Linear Layer 1 | 512 | - |
| Hidden Linear Layer 2 | 16 | - |
| Output Layer | $|A|$ | - |

Table 3: All Hyperparameters for DQN and BCQ [Atari]

| Hyper-parameter | Value |
|---|---|
| Network optimizer | Adam Kingma & Ba (2015) |
| Learning rate | 0.0000625 |
| Adam $\epsilon$ | 0.00015 |
| Discount $\gamma$ | 0.99 |
| Mini-batch size | 32 |
| Target network update frequency | 8k training updates |
| Evaluation $\epsilon$ | 0.001 |
| Threshold $\tau$ (BCQ) | 0.3 |

Table 4: All Hyperparameters for DQN and BCQ [3D Nav]

| Hyper-parameter | Value |
|---|---|
| Network optimizer | Adam Kingma & Ba (2015) |
| Learning rate | 0.0003 |
| Discount $\gamma$ | 0.99/0.95 |
| Mini-batch size | 64 |
| Target network update frequency | 100 training updates |
| Evaluation $\epsilon$ | 0.001 |
| Threshold $\tau$ (BCQ) | 0.3 |

Table 5: All Hyperparameters for $DQN^*$

| Hyper-parameter | Value |
| --- | --- |
| Replay buffer size | 1 million |
| Training frequency | Every 4th time step |
| Warmup time steps | 20k time steps |
| Initial $\epsilon$ | 1.0 |
| Final $\epsilon$ | 0.01 |
| $\epsilon$ decay period | 250k training iterations |

