# OpenReview forum: "DeepAveragers: Offline Reinforcement Learning By Solving Derived Non-Parametric MDPs"
_ICLR.cc/2021/Conference — ICLR 2021 Spotlight_

### Official Review · AnonReviewer1 · 2020-10-19
**A good paper**

**Rating:** 7
**Confidence:** 4

**Review:**

The authors propose a method to construct non-parametric models for offline control based on the averager framework. The authors tested three typical latent representations including random projection, DQN representation, and BCQ representation. KNN is then used to construct the non-parametric model on the latent space. The reward function is augmented with a penalty term penalizing unseen transitions in the dataset. A novel GPU MDP solver based on value iteration is used to solve the non-parametric model. Theoretically, a performance lower bound is derived for the approximate MDP. Empirically, a thorough comparison with baselines is performed in the challenging Atari domains. I particularly like the case study in the 3D navigation domains, which highlights the flexibility of DAC-MDP.

Overall I think the paper makes a good contribution in offline RL. The ideas are intuitive and the empirical results are convincing.

Minor comment:
(1) I can understand building the kNN table is a one time cost in offline setting as the dataset does not change. In the online learning setting, building such a kNN table is usually considered computationally intractable, so I think the paper may benefit from providing some statistics of the run time for compiling such a kNN table for 2.5M Atari frames, as well as the memory cost.
(2) The paper implements a hybrid version of Block Divided Iteration and Result Divided Iteration. I think the paper may benefit from some discussion about the convergence of this hybrid GPU value iteration algorithm.
(3) According to how I understand the paper, a Eucledian distance is used on the latent space. When random projection is used, I can see it can easily project (s, a) together. But when DQN or BCQ representation is used, I don't fully understand how they encode (s, a). In my understanding, the default DQN and BCQ encode only states, so how are actions encoded?
(4) Contrastive learning has recently achieved great success in representation learning, as well as reinforcement learning as auxiliary tasks. I'm wondering if it can help improve the performance of DAC-MDP.

---

> ### Author Response · Authors · 2020-11-24
> **Response to Review**
>
> Thank you for the review and useful comments for improving the paper. We have uploaded an updated paper with a number of revisions. We added a comment with a change log and also reference relevant changes in our response.
>
> *"building such a kNN table is usually considered computationally intractable … run time for compiling such a kNN table for 2.5M Atari frames, as well as the memory cost."* \
> **Re** We have included these details in the current revision. As a preview, the compilation of MDP for the 2.5M dataset takes 1-1.5 hrs and the VI solver takes less than 3 minutes. This is a highly unoptimized CPU implementation of the compilation and could be improved using a GPU enhanced KD Tree approach [1,2]. The compiled MDPs range from 2-6 Gb depending on the action space, which again is not fully optimized in the current implementation. There are many ways to further improve scalability to help address an online setting. Incremental updates are one of the main approaches, which could be effective due to each new data point only influencing nearby parts of the model.
>
> [1] ]Martin, W. W.. *"Implementation of Kd-Trees on the GPU to Achieve Real-Time Graphics Processing."*  (2012). \
> [2] Hu, Linjia, et al. *" Massively parallel KD-tree construction and nearest neighbor search algorithms."* (ISCAS) (2015): 2752-2755.
>
> *"some discussion about the convergence of this hybrid GPU value iteration algorithm."* \
> **Re** We have added a footnote in the discussion of GPU VI.  The implementation variants still simulate the exact value iteration, so it converges to the optimal policy just as VI does.
>
> *"In my understanding, the default DQN and BCQ encode only states, so how are actions encoded?"* \
> **Re** As mentioned in the second paragraph of Section 3.1 the distance function is defined to be infinite for different actions. Hence the distance between (s, a) pairs can be calculated by the euclidean distance between the latent states. We have expanded on this in 3rd paragraph in the preliminaries section.
>
> *"Contrastive learning …  I'm wondering if it can help improve the performance."* \
> **Re** The DAC-MDP framework is agnostic to the approach used to provide the latent representation, so any representation learning approach, including contrastive learning, is worth exploring.

---

### Official Review · AnonReviewer4 · 2020-10-28

**Rating:** 7
**Confidence:** 3

**Review:**

The authors propose to learn a non-parametric MDP model from batch data, which can be solved efficiently using discrete value iteration (by solving for the “core” states which are all the end-states in observed transitions) and which provides a Q-value defined over the full continuous space through a kNN lookup. There is an interesting penalty term when the estimate relies on far-away support data. The value of the optimal policy in the approximated MDP can be bounded, under some smoothness assumptions, in the original MDP.

My main concern about this first contribution is about novelty. The authors have not cited other classic papers on this topic which have similar results, for example: Kernel-Based Reinforcement Learning by Ormoneit and Sen, 2002 (see in particular Section 4 which also mentions how dynamic programming can be run on the “core” states). The authors should discuss the relation to this work and surrounding literature. Follow-up papers like (Tree-Based Batch Mode Reinforcement Learning, Ernst et al. 2005) have been applied to large domains and fitted Q-iteration (FQI) has been studied in many papers afterwards both empirically and theoretically. It would be great if the authors could comment on that.

The second paper contribution is to explore the practicality of this approach using various ways to represent the state, including parametric functions as used in DeepRL. The studied domains are Cartpole, Atari (8 games) and 3D navigation tasks. The empirical results in Cartpole demonstrate the importance of the penalty term to obtain good performance. It would be nice to know the score of the pre-trained Cartpole policy for comparison in Fig 2 (same question for 3D nav domains). In Atari, it appears that the latent representation can be leveraged by the proposed approach to robustly improve the DQN score. In general these experiments are usefully presented, but some questions remains over how other approaches would do in similar settings. For example:
* Instead of using the learned representation outside of the original DeepRL setting, can that extra computation be used instead within the same DQN framework by replaying the same data and updating the network further? (cf paper When to use parametric models in reinforcement learning? by van Hasselt et al. 2019)
* Can LSPI be run using these representations? What are the accuracy/speed trade-offs there?
* What about other FQI-style methods like Extra-trees cited above? Or in general how do other batch RL methods compare?

3D Nav: “ For most offline RL approaches this would require retraining the agent on a modified dataset from scratch.” For this scenario, perhaps only the last layer could be retrained while keeping the same latent representation, which is similar to what is going on here? The DAC-MDP agent is retrained “from scratch” with this different cost function, though perhaps this is much faster in this case.

The 3D nav experiments are interesting, but it’s hard to know what to take away from them. If flexibility of task modifications using the same data batch + learned representation is the main point, it seems important to quantify this further (other approaches can do this zero-shot transfer, but perhaps not as well or quickly?). Also this method would presumably fail as the pre-learned representation becomes less relevant for the modified task.

Overall, the paper is written clearly and it revisits older RL work to demonstrate their practicality combined with modern tools. The experiments do a good job with ablations of the proposed method but without clear baselines to compare to. As mentioned above,  previous work in this area needs to be more thoroughly referenced and discussed, the small “Averagers Framework” Related Section is missing some connections to past work.


Minor points:

* I am not sure k_\pi is defined. Is it the k parameter in Eq 3?
* “eucledian” distance
* Fig 2 legend “parameter”
* Clarify iterations in the Atari setting (frames? steps?)
* “their optimal polcies are visualzed”


---- Post-rebuttal ----

Thank you for addressing my concerns and providing the additional experiments and baseline which I think make the paper stronger. I have updated my score as a result.

---

> ### Author Response · Authors · 2020-11-24
> **Response to Review (Part 2)**
>
> *"perhaps only the last layer could be retrained while keeping the same latent representation, …. The DAC-MDP agent … perhaps this is much faster in this case."* \
> **Re** This is a reasonable idea which we have experimented with for the new revision. We run the baselines for secondary objectives from scratch and discuss the performance and walltime. For 3D nav domains, we find that DAC-MDPs generally perform better and are significantly faster for secondary objectives (25-40x). While it is true that just retraining the last layer is faster, we saw less than 30% wall-time improvement by keeping the previous layers fixed. This is expected as the network is relatively small, to begin with, but we do note that this gain may be more significant in bigger networks. Note that one challenging issue with this type of retraining (and actually the original training) is knowing when to stop the iteration without using online evaluation. Prior work mostly ignores this issue.  Moreover, we note that DQN and BCQ have significantly longer runtimes than our approach in this domain as pointed out in the computational analysis section in the revision.
>
> *"The 3D nav experiments are interesting, but it’s hard to know what to take away from them. … important to quantify this further (other approaches can do this zero-shot transfer, but perhaps not as well or quickly?)"* \
> **Re** Our main objective for 3D navigation was to demonstrate the flexibility of our approach to rapidly adapt to different objectives, action sets, and stochasticity. However, in the current revision, we have expanded these results to also include comparisons to the baselines (DQN and BCQ) in terms of both performance and learning time. As you suspected, baseline approaches cannot do this as well or quickly. We have added them in the computational analysis section and appendix A.6
>
> *"presumably fail as the pre-learned representation becomes less relevant for the modified task."* \
> **Re** Agreed. The model-based transfer is only appropriate if the model is relevant to the new task.
>
> **Minor points:**
>
> *"I am not sure k_pi is defined. Is it the k parameter in Eq 3?"* \
> **Re** Yes it is the same k used in Equation 3 but for policy computation, as explained in the practical issues (section 3.1, further elaborated in the last paragraph of Appendix A.1.
>
> Other minor issues have also been fixed in the current revision.

---

> ### Author Response · Authors · 2020-11-24
> **Response to Review (Part 1)**
>
> Thank you for the review and useful comments for improving the paper. We have uploaded an updated paper with a number of revisions. We added a comment with a change log and also reference relevant changes in our response.
>
> *"The authors have not cited other classic papers on this topic which have similar results"* \
> **Re** It may not be accurate to say that Ormoneit and Sen, 2002 have similar results. The conceptual ideas are similar, but they did not include any experiments, let alone demonstrate successful results on image-based RL benchmarks.
> We agree that it is appropriate to add a citation to (Ormoneit and Sen, 2002) and other related KBRL work. We have done so in the current revision. We chose to cite Gordon (1995) for this concept (second paragraph of Section 3) as this was the original source that we drew inspiration from.
>
> *"My main concern about this first contribution is about novelty."* \
> **Re** It is true that there is a long history of prior work that builds on optimal tabular solvers. However, we are unaware of any prior work that demonstrated positive results on challenging image-based Deep RL benchmarks such as Atari. We would be interested to see pointers if you know of any.
>
> The closest prior demonstration that we are aware of is VaST (Corneil et al., 2018). However, in that paper, the authors explicitly say that they were not able to get VaST to work for Atari games other than the deterministic version of Pong (we are using stochastic Atari in our experiments).
>
> While, in hindsight, our approach seems straightforward, it was not at all clear at the start that it had a chance of working---as reviewer 1 mentioned it is surprising. The surprise is a result of significant work to identify the key ingredients needed for success (pessimistic costs combined with the averagers framework).
>
> We hope that you will appreciate the value of demonstrating the potential of a very different approach compared to the norm in DeepRL. Such a demonstration can inspire others to investigate this style of under-explored approach, with hard-to-predict implications.
>
> *"The authors should discuss the relation to this work and surrounding literature … (Ernst et al. 2005) have been applied to large domains ... It would be great if the authors could comment on that."* \
> **Re** FQI is more appropriate to compare to model-free approaches such as DQN and variants. We are taking a model-based approach here that uses optimal tabular planning. We have never seen a tree-based method applied to image-based benchmarks such as Atari. We would be interested in a pointer. Currently, we are focusing the related work on the most closely related model-based RL approaches. In particular, there isn't enough space to cover the huge number of model-free approaches in the literature. Having said that, we have updated the related work to cover kernel regressor adaptations of FQI such as KBRL.
>
> *"score of the pre-trained Cartpole policy for comparison"* \
> **Re** We have included this in the current revision. As a preview, in Cartpole, the performance was 20, 140, and 500 for randombag, mixed, and optimal respectively. For 3D Nav, the performance was -9.84 for the simple/boxpillar room and -25 for the Tunnel room.
>
> *"extra computation be used instead within the same DQN framework by replaying the same data?"* \
> **Re** In our experience, running DQN or BCQ longer does not significantly improve performance. Since the size of the dataset is fixed in offline RL, this is not surprising since there is only so much an algorithm can pull out of the data. One can see in Figure 3 that the approaches appear to saturate and sometimes even get worse with more computation.
>
> *"Can LSPI be run using these representations? What are the accuracy/speed trade-offs there?"* \
> **Re** This will be an interesting comparison to make in the future. LSPI can in concept be run on the latent representations we used for DAC-MDPs. We currently don’t have a scalable enough version of LSPI and it is unlikely that we can do so for any upcoming revisions. We note that LSPI will be limited to Q-functions that are linear in the latent features, while the DAC-MDP approach can produce non-linear Q-functions due to its non-parametric nature.
>
> *"What about other FQI-style methods like Extra-trees cited above? Or in general, how do other batch RL methods compare?"* \
> **Re** We don’t know how to apply FQI-style methods based on trees to image-based domains. It seems unlikely that giving pixels directly to trees will result in anything reasonable. We have compared to DQN and BCQ, noting that BCQ is a recent competitive offline approach.

---

### Official Review · AnonReviewer2 · 2020-11-02
**Reinforcement learning for scaling data-driven dialog policies**

**Rating:** 7
**Confidence:** 3

**Review:**

Summary
=========
Authors introduced the Deep Averagers with Costs MDP (DAC-MDP) as an offline approach for solving MDPs
They addressed the following planning challenges: difference in representation used by model and planner, and 2) planners exploit inaccuracies of models
To address (1) authors relied on a simple tabular representation. To address (2), the adopted pessimism in the face of uncertainty by encouraging the agent to focus on known areas.

Pros / Cons
[+] Theoretical analysis of DAC-MDP

[+] Empirical results on toy domain (Cart pole), Atari and 3D first-person navigation task

[+] The idea is very simple

[-] Some of the decisions made by the authors felt arbitrary (See details/questions)

[-] Authors used 3 seeds for empirical results which seems too small. Moreover, some plots lack statistical significance.

[-] As with any planning approach, these algorithms are expected to perform better than their counter part without planning.
However, there is no free lunch and the boost will come at the expense of computation. Currently the paper does not discuss that critical angle.

[-] The main idea does not seem very novel (see below)

Questions:
==========
Eqn. 2: Why not scale the rewards based on their distance to (s,a) instead of having the distance as a penalty?

Page 6:
	- Atari domain has deterministic transition. How well this approach works for stochastic transitions?
	- The entire 100K iteration protocol was repeated 3 times => Is 3 times enough to provide statistical significance?
	- Figures 4 investigates the performance at the final iteration for different values of Ne. For Ne = 1 we use k=5,kπ =11,C=1 and
for Ne =20 we expand on the parameter set used for Ne=6. => How did you select these parameters?

Figure 5: Where are the standard errors? Are all differences significant? Again you should also discuss wall clock time.

Page 8: The idea of moving non-core states to core states for planning using a metric is not new (e.g. [1])

Generic: I am not sure how easily this approach can scale to large stochastic domains with continuous state space. Was the 3D first-person navigation task included stochasticity in the transition function?

Details
=========
P3: "non-core state immediately transitions to the core for any action" => How do you calculate the corresponding core state? Is it the closest using KNN?
Figure 2: Where are the error bars?
Figure 3: What about computation time?

[1] J. Joseph, A. Geramifard, W. Roberts, J. How and N. Roy,  “Reinforcement Learning with Misspecified Model Classes”, IEEE International Conference on Robotics and Automation (ICRA), 2013


POST-REBUTTAL
==============
"First, we would argue that there are not many examples of planning approaches outperforming non-planning approaches when using imperfect learned models. Providing such a demonstration in challenging image-based benchmarks is one of the contributions of this work."
- I disagree with authors. The idea of planning using imperfect models have been explored as early as 1998 (e.g. Dyna algorithm in Reinforcement Learning: An Introduction by Sutton and Barto).

"computational analysis"
- Thank you for adding this section. It addressed my concern.

"non-determinism"
- Thank you for adding more clarity. It addressed my concern.

"Statistical significance"
- Great to see the extra runs. Please include the bars for Figures 5,7 in the final submission.

---

> ### Author Response · Authors · 2020-11-24
> **Response to Review (Part 2)**
>
> *"moving non-core states to core states for planning using a metric is not new (e.g. [1**)”* \
> **Re** Agreed. We do not claim this as a novel contribution. In the second paragraph of Section 3, we cite Gordon (1995) as the original motivation for this (i.e. the original averagers framework). Our main contribution lies in the identification/integration of the costs framework to the non-parametric MDPs, which are essential to make the framework work.  Moreover, we are not aware of prior work that demonstrates this idea of challenging image-based RL domains such as Atari.
>
> *"Generic: … scale to large stochastic domains with continuous state space... "* \
> **Re** You can see this scaling in the Atari experiments for image-based observations. The Atari experiments are stochastic (using sticky actions), which is detailed in Appendix B.1. We have made this explicit in the current revision. The 3D navigation domain is deterministic following prior work.
>
> **Details**
>
> *"How do you calculate the corresponding core state? Is it the closest using KNN?"* \
> **Re** Yes, as detailed in Equation 3, the Q-values are computed based on KNN. This follows directly from the formalism of the averagers framework.

---

> ### Author Response · Authors · 2020-11-24
> **Response to Review (Part 1)**
>
> Thank you for the review and useful comments for improving the paper. We have uploaded an updated paper with a number of revisions. We added a comment with a change log and also reference relevant changes in our response.
>
> *"As with any planning approach, these algorithms are expected to perform better than their counterpart without planning. However, there is no free lunch and the boost will come at the expense of computation. Currently, the paper does not discuss that critical angle."* \
> **Re** First, we would argue that there are not many examples of planning approaches outperforming non-planning approaches when using imperfect learned models. Providing such a demonstration in challenging image-based benchmarks is one of the contributions of this work.
>
> We agree that more detail about computation time will improve the paper. We have added this additional discussion to the current revision under the "computational analysis” section at the end of experiments. As a preview, for the 100k Atari results, it takes less than 10 minutes to both build and solve a DAC-MDP. The vast majority of the computation comes from the KNN computations. This has a throughput of 5000 lookups/seconds currently. It is a single-threaded CPU implementation that is not very optimized. This can be optimized further using GPU optimized KD-Trees. The GPU VI solver comfortably solves any size MDP (up to millions) within minutes. (less than a minute for size of 100k). In the current revision, we have compared this with the time used to train the model-free baselines for both primary and secondary objectives.  (We are able to get 25-40x improvement in wall time for secondary objectives)
>
> *"The main idea does not seem very novel”* \
> **Re** It is true that there is a long history of prior work that builds on optimal tabular solvers. However, we are unaware of any prior work that demonstrated positive results on challenging image-based Deep RL benchmarks such as Atari. We would be interested to see pointers if you know of any.
>
> The closest prior demonstration that we are aware of is VaST (Corneil et al., 2018). However, in that paper, the authors explicitly say that they were not able to get VaST to work for Atari games other than the deterministic version of Pong (we are using stochastic Atari in our experiments).
>
> While, in hindsight, our approach seems straightforward, it was not at all clear at the start that it had a chance of working---as reviewer 1 mentioned it is surprising. The surprise is a result of significant work to identify the key ingredients needed for success (pessimistic costs combined with the averagers framework).
>
> We hope that you will appreciate the value of demonstrating the potential of a very different approach compared to the norm in DeepRL. Such a demonstration can inspire others to investigate this style of under-explored approach, with hard-to-predict implications.
>
> *"Why not scale the rewards based on their distance to (s, a)?”* \
> **Re** The current additive penalty approach is motivated by the theoretical analysis. An additive penalty is a natural form when starting from an assumption of Lipshitz continuity. Perhaps there is an analysis that would result in a scaled penalty---we would be interested to see what that would look like.
>
> *"Atari domain has a deterministic transition. How well this approach works for stochastic transitions?"* \
> **Re** This is incorrect. The experiments are for stochastic Atari with sticky actions enabled. This is detailed in Appendix B.1. We have made this explicit in the main text of the current revision.
>
> *"Is 3 times enough to provide statistical significance?", "Figure 5: Where are the standard errors?"* \
> **Re** We are currently showing 95% confidence intervals based on the 3 seeds. Our pre-submission computational constraints limited us to 3 seeds. We have run the experiments for 2 additional seeds for a total of 5 seeds and have updated the results in the current revision. The intervals shrink a bit with no qualitative differences in performances between methods.
>
> *"Figure 4 investigates the performance at the final iteration ... we expand on the parameter set used for Ne=6 … How did you select these parameters? "* \
> **Re** As described in Appendix A.1 we vary the cost parameter C orders of magnitude above and below the value selected by our heuristic (C=1 in this case). So, for Ne=20 we select C from {1,10,100,1k,1M} and then consider one additional option for k_pi choosing from {5,11,31,51}. The combinations of these form the parameter sets used. We have clarified this in the current revision.

---

### Official Review · AnonReviewer3 · 2020-11-03
**Better batch RL via penalizing poorly-understood state-actions**

**Rating:** 7
**Confidence:** 3

**Review:**

This paper proposes DeepAveragers: an adaptation of the "averagers" framework to the Deep RL setting.
Approximate dynamic programming is performed over a batch of observed (s,a,r,s') except with a distance penalty applied to states that are not close to those observed in the training data.
The authors show that this formalism can lead to better performance than learning without penalty, and also that better representations of "state similarity" lead this method to perform better in aggregate.

There are several things to like about this paper:
- The problem of "batch RL" is timely as people look to apply RL techniques in real world problems, where learning from tabula rasa may be prohibitive.
- The proposed method appears sound, general and well thought through.
- The experimental results provide a clear progression from theory -> didactic examples -> deep RL, and the approach performs well across the board.
- The paper is overall well-written and easy to follow.

However, there are a few places where it might be improved:
- The discussion and results of "3D navigation" are not well-presented... large paragraphs of texts with relatively arbitrary numbers (0.98 vs 0.96) do not make a convincing presentation. Can these be displayed better visually?
- I'm not convinced enough connection is given between the "averager" framework and the many other approaches to batch RL, which all essentially seem to work via a similar manner: give penalties to the state-actions that are poorly understood.
- I think that some of the GPU-specific discussion is a bit distracting/incidental to the core work of the apper.


Overall, I like the paper and think it seems like a strong contribution.
I'm slightly worried that the comparison to other related work is insufficient, as I am not well-versed in the area.
However, from what I can see it looks like it is suitable for the conference.

---

> ### Author Response · Authors · 2020-11-24
> **Response to Review**
>
>
>
> Thank you for the review and useful comments for improving the paper. We have uploaded an updated paper with a number of revisions. We added a comment with a change log and also reference relevant changes in our response.
>
> *Concern: "results of ‘3D navigation’ are not well-presented"* \
> **Re:** Our main objective for 3D navigation was to demonstrate the flexibility of our approach to rapidly adapt to different objectives, action sets, and stochasticity. However, in the current revision, we have expanded these results to also include comparisons to the baselines (DQN and BCQ) in terms of both performance and learning time. The time for our approach to adapt the policy will be much faster than the time to retrain DQN or BCQ.
>
> *"Concern: Not enough connections given to "Averager framework" and many other approaches to batch RL … seem to work via a similar manner: give penalties to the state-actions that are poorly understood."* \
> **Re:** While they do look similar we are not aware of any prior or recent approaches that successfully combine optimal tabular planning with penalty methods. We would be interested in any pointers. Averagers framework is the essence of many approaches. In related work, under the heading "Pessimism for Offline Model-Based RL" we covered this with a focus on model-based methods since we are focusing on model-based techniques. Moreover, we have expanded on this to include other approaches using the additional space in the current revision.
>
> *"GPU-specific discussion is a bit distracting/incidental."* \
> **Re:** Other than the 3-line discussion on the efficacy of the GPU implementation in the practical issues section, most of the GPU discussion is in the appendix. We opted to replace many mentions of value iteration with "GPU optimized value iteration" as we feel that it is worth a brief mention in the main paper since scalability is one of the first questions someone might have about this style of approach. The GPU implementation is an important part of scalability.

---

### Official Review · AnonReviewer5 · 2020-11-04
**Good results for a simple approach, but low novelty & a lot of minor issues**

**Rating:** 6
**Confidence:** 4

**Review:**

The authors present a nearest neighbour method for learning a model offline from the statistics of the given data set. Representations are provided from other off-policy deep RL methods. Value iteration is used on top of the model to learn the final policy. The algorithm is tested in several Atari games over two data set sizes.

Strengths:
- Overall, the proposed method is straightforward, intuitive, and outperforms baselines on a challenging task. The results are somewhat surprising given the simplicity of the approach.
- With a few exceptions (discussed below), the presentation quality is high. Figures are clear/readable. Writing is clear.
- With a few easily fixed exceptions (discussed below), reproducibility is high. The appendix is detailed & includes additional experiments. Code is provided. Most algorithmic and experimental details are provided.

Weaknesses:
- Experimental results suggest the method requires a representation derived from other off-policy deep RL algorithms, meaning running the proposed method also requires running another deep RL method to simply learn the representation. At the same time, the performance gains seem incremental for a lot of additional complexity.
- I do feel like there is a contribution here, but novelty is not that high compared to other methods which build on tabular RL.
- Although I can (personally) infer the purpose of $d(s,a,s_i,a_i)$ in the reward function from related work in the field, the use of $d$ is not well motivated by the authors. At the same time, both the choice of $C$ and $d$ would seem to be important hyper-parameters relative to the scale of the reward function. A choice which is difficult to make without interacting with the environment. Furthermore, I was unable to find the description of the distance $d$ anywhere in the main body or appendix.
- The related work is missing discussion on recent offline RL methods such as [1,2,3,4].
- Baselines are limited to only two methods. Possible additions [1,3]. Additionally, although its nice that additional experiments were includes the gym mini-world experiments don't add much to the paper without baselines.
- Reproducibility Issues:
    - The use of $N_e$ is unusual. Its role is not well described in Section 2 without reading the appendix.
    - I was unable to find the number of seeds used in each experiment. Although error bars are provided in some figures, I was unable to find what quantity these error bars were describing.
    - As previously mentioned, I could not find which distance function $d$ was used in the experiments.

Minor Clarity Issues:
- Eqn (1) and (3) describe the bellman equation over Q functions, however the method relies on state value functions $V$ which are learned with a transition function (such as described in Pseudocode 1 in Appendix A.6). Fixing this along with a clearer description of the 1-step kNN look ahead step would improve clarity.
- Typo: Figure 2: "Datset" version.
- Typo: A.2. "Theoritial Proofs".

Recommendation:

Overall, I think the results of the method are interesting, even if incremental. In its current state I would favor rejection, but I believe this would be a solid paper with enough corrections to the weaknesses I've listed. Either during the rebuttal phase of this conference or as a submission to a future conference.

References:
- [1] Agarwal, Rishabh, et al. "An Optimistic Perspective on Offline Reinforcement Learning." 2020.
- [2] Fujimoto, Scott, et al. "Off-policy deep reinforcement learning without exploration." 2019.
- [3] Wang, Ziyu, et al. "Critic regularized regression." 2020.
- [4] Levine, Sergey, et al. "Offline reinforcement learning: Tutorial, review, and perspectives on open problems." 2020.

**Post-Rebuttal**

The authors have done well with their additional page, and many of the concerns I had have been dealt with the latest iteration of the paper. I have increased my score.

Re: Motivation for c and d. The current *objective* for c and d is described in the paper, but the motivation is not. I hope the authors add to this discussion in the next iteration.

---

> ### Author Response · Authors · 2020-11-24
> **Response to Review (Part 2)**
>
> *"c and d … seem to be important hyper-parameters … choice which is difficult to make without interacting with the environment. "* \
> **Re:** The amount of online interaction, if any, according to our evaluation protocol is specified by N_e (the number of policies that can be evaluated online). Note that most prior work on offline RL doesn’t even specify such a value and uses arbitrary amounts of the online experience for parameter search---whereas we are being very explicit about the online experience.
>
> As described in the appendix "Hyper Parameter Selection" our heuristic for selecting c is to set it to be the same magnitude as the reward signal. If we have more than one online evaluation available (N_e > 1) we use values that are orders of magnitude above and below the heuristic value. However, it is important to note that our approach while benefiting from hyperparameter sweep is not highly sensitive to it. Even for a single heuristic hyperparameter setting Ne=1 our approach does non-trivially well and beats DQN baselines. Moreover, the gains from Ne=6 to Ne=20 is not highly significant.
>
>
> *" related work is missing discussion on recent offline RL methods "* \
> **Re:** We have added [1,2,3] to our related work section in the current revision. We focused related work primarily on the most related work for space reasons. Note that [4] is a survey that we have added when we first introduce the concept of offline RL.
>
>
> *"Baselines are limited … Possible additions [1,3]... gym mini-world experiments don't add much to the paper without baselines. "* \
> **Re:** The implementation of [3] is not available to our knowledge and the baselines provided in [1] were shown to be comparable to BCQ in Gulcehre et al. (2020) [RL Unplugged] over Atari Datasets and we already compare to BCQ.
>
> We are currently running DQN and BCQ on the 3D gridworld environment and will include those results in the upcoming revision. Note that DQN and BCQ have significantly longer runtimes than our approach in this domain as pointed in the computational analysis section in the revision. (we use random-projection for a representation in the domain).
>
> **Reproducibility issues:**
>
> *"The use of Ne is unusual."* \
> **Re:** We argue that the Ne protocol is not unusual, but discussing it explicitly in a paper is. Existing offline RL approaches also have hyperparameters (network architecture, optimizer, learning rates, etc) and often report results for the best performing hyperparameters as measured by online evaluations (effectively using a very large, unspecified Ne). Our intent is to introduce an evaluation protocol that forces clarity about the amount of online experience used for tuning via Ne (as outlined at the end of Section 2). We are open to alternative evaluation protocols that address this issue.
>
> *"number of seeds, … what quantity these error bars were describing. "* \
> **Re:** The results in the paper used 3 seeds and the error bars are 95% confidence intervals. These were all we could produce given the computation constraints before submission. We have run the experiments for 2 additional seeds and have updated the results in the current revision. The intervals shrink a bit with no qualitative differences in performances between methods.
>
> *"which distance function d"* \
> **Re:** As mentioned in the paper (Section 3.1 Second paragraph), the distance between state-action pairs with different actions is considered to be infinite. Otherwise, the distance is the Euclidean distance between the latent state representations. We have clarified this in the current revision.
>
> *"a clearer description of the 1-step kNN look ahead step would improve clarity."* \
> **Re:** We have added a footnote regarding the 1-step lookup computation of Q values in the current revision.

---

> ### Author Response · Authors · 2020-11-24
> **Response to Review (Part 1)**
>
> Thank you for the review and useful comments for improving the paper.
> We have uploaded an updated paper with a number of revisions. We added a comment with a change log and also reference relevant changes in our response.
>
> *Concern 1: "The results suggest that the method requires learned representations from other off-policy deep RL algorithms for strong performance."* \
> **Re:** Our approach is agnostic to the source of the representation and results in non-trivial improvements in performance and flexibility when operating on deep RL representations.
> In Figure 5, we show results using a random representation baseline (purple bars). This representation is competitive with the off-policy RL versions for many tasks -- outperforming standard DQN in pong, space invaders, bowling, freeway, and amidar.  Being representation agnostic, our approach may also benefit from advances in representation learning including contrastive methods. Exploring a significant variety of these is outside the scope of this paper.
> Further, even when using representation from off-policy deep RL algorithms, our approach offers a number of advantages by creating an explicit MDP and optimally solving it. We see quantitative improvements (see next concern) and we can alter policy parameters/rewards without needing to retrain the representation as demonstrated in the 3D navigation setting.
>
>
> *Concern 2: "performance gains seem incremental for a lot of additional complexity."* \
> **Re:** We politely disagree on both points.
> Complexity: Our approach is simple to apply (as R3 notes as a strength) -- value iteration on a tabular MDP is well-developed and producing a DAC-MDP from a finite dataset is straight-forward. It is this simplicity that we find meritorious in contrast to the complexity of SGD based off-policy model learning. Computationally, our (largely unoptimized) implementation takes under 90 minutes for the 2.5 million datasets -- only 3 minutes of which is value iteration.
>
> Gains: We believe our gains are not incremental. Applying our approach to DQN leads to improvements of 300% normalized score on average for stochastic Atari tasks. (60% for BCQ).  This is with no additional data and on the same representations -- our approach simply solves for a better policy. If we have misunderstood the concern, we are happy to receive further comments.
>
> *Concern 3: "there is a contribution here, but novelty is not that high"* \
> **Re:** It is true that there is a long history of prior work that builds on optimal tabular solvers. However, we are unaware of any prior work that demonstrated positive results on challenging image-based Deep RL benchmarks such as Atari. We would be interested to see pointers if you know of any.
>
> The closest prior demonstration that we are aware of is VaST (Corneil et al., 2018). However, in that paper, the authors explicitly say that they were not able to get VaST to work for Atari games other than the deterministic version of Pong (we are using stochastic Atari in our experiments).
> While, in hindsight, our approach seems straightforward, it was not at all clear at the start that it had a chance of working---as you mentioned it is surprising. The surprise is a result of significant work to identify the key ingredients needed for success (pessimistic costs combined with the averagers framework).
>
> We hope that you will appreciate the value of demonstrating the potential of a very different approach compared to the norm in DeepRL. Such a demonstration can inspire others to investigate this style of under-explored approach, with hard-to-predict implications.
>
> *Concern 4: "the purpose of distance(s, a,s’, a’) is not well motivated by the authors."* \
> **Re:** The current motivation for the adjusted reward (i.e. using c and d) is in the paragraph before 3.1 and the second to last paragraph of the introduction. Your intuition about the motivation is correction. We agree that increasing this discussion in the final version will improve the paper. This will be in the next revision.

---

### Author Response · Authors · 2020-11-24
**Changes Log**


We have made the following changes to the paper:
**Major Changes:**
- [C1] Added Baseline experiments for 3D navigation Domain.  \< Appendix
   A.6, pointers added in the 3D nav experiments section of the main paper \>
- [C2] Added Discussion on computational aspects of the approach.   \< subsection Computational Time Analysis  under
   Experiments section \>
 - [C3] Updated Atari experiments for 5 seeds
   instead of 3. < Figure 3 , Figure 4 >
 - [C4] Expanded on Averagers line of work to include works like KBRL(Ormoneit and Sen, 2002). < “Averagers
   Framework” subsection of Related Work >
 - [C5] Added a discussion on more recent model-free offline Reinforcement Learning approaches. \<
   “Model free RL for Offline settings” sub-section of Related Work >


**Minor Changes:**
- [c1] Added 3D Navigation Domain Visualization in main paper. \<  Figure 6 >
- [c2] Briefly expanded on distance function. \< second paragraph of section 3.1 >
- [c3] Added error bars description in figures. \< all figures >
- [c4] Added a footnote regarding 1-step lookup computation of Q values. \< footnote 1, page 4>
- [c5] Made the stochasticity of Atari games explicit. \< first paragraph Atari experiments section>
- [c6] Added the parameter choices for Ne=20. \< second last paragraph of Atari experiments section>
- [c7] Added scores for pre-trained policies for both Atari and 3D nav domains. \< footnote 2, page 5>
- [cX] All minor typos fixed.

---

### Decision · Program_Chairs · 2021-01-07
**Final Decision**

**Decision:**

Accept (Spotlight)

**Comment:**

This paper proposes an algorithm for offline RL, that consists in solving a finite MDP derived from a fixed batch of transitions.
The initial reviews were overall positive, and the concerns raised at this stage were nicely addressed by the rebuttal and the revision from the authors.
The final discussion led to the consensus that this paper should be accepted at ICLR.